# NeST: nested hierarchical structure identification in spatial transcriptomic data

Benjamin L. Walker [ORCID][1,2] & Qing Nie [ORCID][1,2,3] ✉

Spatial gene expression in tissue is characterized by regions in which particular genes are enriched or depleted. Frequently, these regions contain nested inside them subregions with distinct expression patterns. Segmentation methods in spatial transcriptomic (ST) data extract disjoint regions maximizing similarity over the greatest number of genes, typically on a particular spatial scale, thus lacking the ability to find region-within-region structure. We present NeST, which extracts spatial structure through coexpression hotspots —regions exhibiting localized spatial coexpression of some set of genes. Coexpression hotspots identify structure on any spatial scale, over any possible subset of genes, and are highly explainable. NeST also performs spatial analysis of cell-cell interactions via ligand-receptor, identifying active areas de novo without restriction of cell type or other groupings, in both two and three dimensions. Through application on ST datasets of varying type and resolution, we demonstrate the ability of NeST to reveal a new level of biological structure.

Spatial transcriptomic (ST) data provides the ability to measure gene expression from cells in tissue while preserving spatial information, allowing insight into the spatial structure of tissue. A variety of ST data collection methods exist, varying in genome coverage, spatial resolution, and capture efficiency, including Visium[1], Slide-seq[2,3], and other alternatives[4,5] which mark locations using spatially-identified barcodes; and multiplexed in-situ hybridization (ISH) imaging based methods[6–11], which through single-molecule imaging measure gene expression levels at single-cell or subcellular resolution.

ST data can be used to understand how groups of cells work together in tissue to perform various biological functions. These groups can exist on drastically different scales: they may contain only a handful of cells, or thousands or millions; they may be groups of cells of the same cell type, or a mixture of multiple different cell types; they may be characterized by the shared expression of only one or a few genes, or thousands. Additionally, the organization of cells in tissue may exhibit a nested hierarchy, where a large structure or region of tissue contains subregions that themselves also have distinct biological meaning and characteristic gene expression patterns, such as structures in the brain built from a collection of internal layers[12,13]. This

region-within-region organization can also be viewed as the spatial analog of identifying subpopulations within a cell type in single-cell RNA-seq analysis, such as studied in immune cells[14,15], stem cells[16], cancer cells[17–19], fibroblasts[20], and for cell activation[21].

Current segmentation methods for ST data analysis divide cells or spots in the dataset into regions such as to maximize some measure of within-region similarity[22]. While this bears similarity to clustering in scRNA-seq data, the desire to obtain spatially coherent regions motivates inclusion of spatial information into the segmentation process. Approaches for this task include expectation-maximization with a Hidden Markov Random Field prior[23,24], fully Bayesian clustering with hyperresolution enhancement[25], and empirical Bayesian clustering[26]. Alternatively, many methods utilize graph neural network (GNN) approaches. SpaGCN[27] integrates segmentation with identification of spatially variables genes. SEDR[28] jointly optimizes two autoencoders, one for expression and one for spatial information. SCAN-IT[29] uses a Deep Graph Infomax[30] (DGI) framework for embeddings. STAGATE[31] applies graph attention (GAT) learning on a cell-type aware network. stMVC[32] uses a multi-view semi-supervised GAT framework combining expression and histological information. SpaceFlow[33] combines a DGI

[1]The NSF-Simons Center for Multiscale Cell Fate Research, University of California Irvine, Irvine, CA 92627, USA. [2]Department of Mathematics, University of California Irvine, Irvine, CA 92627, USA. [3]Department of Developmental and Cell Biology, University of California Irvine, Irvine, CA 92627, USA. ✉e-mail: qnie@uci.edu

framework with spatial regularization of the latent space. An alternative approach is considered in Multilayer[34], in which areas of enriched activity are computed independently for each gene before being combined into a segmentation. However, these methods share a common output data modality: a partition of cells into disjoint spatial regions, limiting the output to representing only one mode of spatial variation that covers the greatest possible number of genes. As a result, those methods are unable to capture multiscale region-within-region structure. In contrast, the hierarchical structure produced with standard agglomerative hierarchical clustering methods, created by repeatedly merging smaller clusters into larger, contains clusters that do not represent spatially localized regions, or fail to combine spatially adjacent and transcriptionally similar cells.

Additionally, most methods require tuning of the spatial scale at which regions are detected, such as by choosing a number of regions, creating challenges when the scale of structure is not known or varies in space. Non-segmentation approaches to analyzing spatial structure in expression include Node-centric Expression Modeling[35], which applies GNNs and variance attribution to identify spatial relationships in gene expression and cell communication, and DIALOGUE[36], which identifies multi-cellular programs, coordinated functional expression patterns dependent on cell type, but do not produce representations of structure in terms of contiguous spatial domains.

We introduce NeST, a method which identifies nested hierarchical structure in ST data through finding coexpression hotspots – representations of spatially localized areas that coexpress a collection of genes. Our method efficiently performs simultaneous searches for coexpression over every possible subset of genes and every spatially contiguous subset of spots, allowing it to operate at multiple scales in both space and number of genes while also capturing nested or overlapping structure in space and identify structures in tissue with no prior knowledge of relevant genes or spatial scale. By applying a spatial diffusion model, NeST is able to identify regions of tissue active in cell-cell interactions (CCI) without being constrained by fixed groupings of cells, such as by cell type. Through application to six ST datasets varying in modality and spatial resolution, we demonstrate the ability of NeST to uncover nested and multiscale biological structure, and to identify spatially localized CCI activity in both two and three dimensions. We further apply downstream analysis and visualization tools to show the localized areas or genes of particular interest, and to capture biological relationships and differences between spatial expression patterns of genes.

## Results

### Identifying nested, hierarchical structure with NeST
NeST is designed to work with ST datasets on any spatial resolution, covering anywhere from 100–20,000 genes, especially those with nested structure, which is illustrated with a Slideseq dataset of the hippocampus[2,3] (Fig. 1a). The hippocampal formation involves four main sub-regions: the CA1, CA2, CA3 regions, and the dentate gyrus (DG). Given an ST dataset, NeST identifies coexpression hotspots (CH), contiguous regions in which some subset of genes are highly expressed (Fig. 1b). Coexpression hotspots are scale-free and may contain arbitrarily few or many spots, and arbitrarily few or many genes, without requiring choice of a preferred spatial scale. Furthermore, as they can overlap, coexpression hotspots have the power to represent the nested structure or other overlapping structures that commonly occur in ST data.

NeST works by first computing single-gene hotspots, localized areas where a particular gene is enriched in expression, for each gene up to full transcriptome coverage. Hotspots are identified by binarizing gene expression using Otsu's algorithm and applying DBscan clustering, producing a separate hotspot for each spatially-dense group of high-expression cells (Fig. 1c1). Then, a hotspot network is constructed, where each hotspot is a node and edges connect hotspots

with a similar shape and location, computed using Jaccard similarity (Fig. 1c2). Finally, communities are extracted from this network, representing groups of highly similar single-gene hotspots, and each group is combined into a single coexpression hotspot (Fig. 1c3, see Methods for details). The largest coexpression hotspot in terms of number of spots/cells is labeled coexpression hotspot 0 (CH0) and further coexpression hotspots are numbered in decreasing order of size. In contrast to a segmentation, any single cell in NeST may be contained in one coexpression hotspot, multiple, or none at all.

By applying a spatial diffusion to expression of ligand genes and then combining with receptor gene expression, NeST computes spatial cell-cell interaction (CCI) hotspots, localized areas in space in which many cells are receiving the effect of a ligand-receptor interaction. Then, the same pipeline is applied to produce coexpression hotspots that relate functional CCI activity to gene activity localized in the same area, as well as perform differential expression and other downstream analysis of the identified functional regions. When three-dimensional spatial data across multiple layers is available, NeST applies a full 3D diffusion model and thereby can compute 3D CCI, including between different layers, producing fully 3D hotspots (Fig. 1d). NeST also contains a variety of downstream analysis tools designed to work with the hotspot framework, such as differential expression analysis and identification of marker genes and decomposition of single-cell expression in terms of coexpression patterns (Fig. 1d).

### Nested hierarchical structure in the hippocampus
The coexpression hotspots identified by NeST accurately capture both the full hippocampal structure, as well as the four subregions within (Fig. 2a). The CA2 region is particularly difficult to identify due to its small size and similarity to CA3, but the coexpression framework allows it to be detected by NeST.

NeST represents the hierarchical relationship among hotspots as a tree, with the smallest hotspot that contains over 75% of another hotspot as the parent of that hotspot. We identify hierarchical marker genes, differentially expressed genes in a region that are enriched relative to parent, sibling, and child coexpression hotspots in the hierarchical structure, but not necessarily relative to coexpression hotspots elsewhere in the tissue (see Methods for details). We find that many NeST marker genes agree with marker genes from a hippocampal marker gene database[37] (Fig. 2b), and all of the NeST marker genes clearly agree with the hierarchical structure (Supplementary Fig. 1a–e, genes for remaining hotspots in Supplementary Fig. 2a–n). We note that such marker genes may be restricted to a single region or expressed in multiple – for example, Neurod6 is expressed in all regions except the DG, and Ncald is expressed in all regions except CA1. Visualizing all marker genes in a heatmap, we see that while almost all CA3 genes have some expression in the CA2 region, the CA2 region has a number of exclusive marker genes (Fig. 2c), and the DG has the most distinct expression profile. Ultimately, the spatial localization inherent to coexpression hotspots filters out cells with similar expression but in a different location (c.f. original annotation of CA1/CA2/CA3 cells in Supplementary Fig. 1fg), allowing for downstream analysis free from spurious inclusion of such cells.

To better understand the presence of nested structure in this dataset, we introduce the nested structure plot, which shows all coexpression hotspots arranged in layers, representing successively finer-scale structure, showing the presence of two layers of structure in the hippocampus, and one layer elsewhere (Fig. 2d). Intuitively, this indicates the presence of up to two layers of overlapping coexpression hotspots within the hippocampal structure, and no overlap elsewhere (see Fig. 1b and all coexpression hotspots in Supplementary Fig. 3). Because both layers are biologically meaningful and clearly substantiated by marker genes, this underscores the value of the coexpression hotspot framework that is able to capture all layers of structure, over a simple segmentation.

**a**   **Input**: ST Dataset

Hippocampal Structure
(black outline)

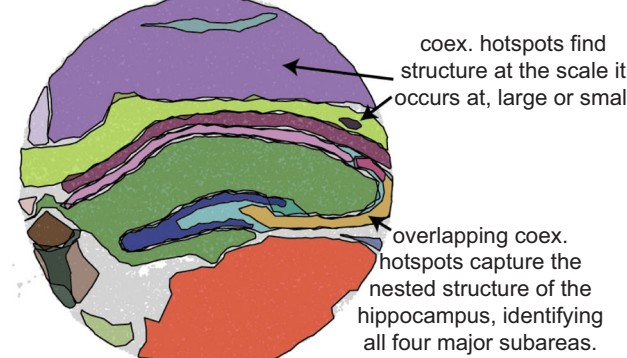

nested structure: large structure (HPC)
contains within meaningful substructures

**b**   **Output: Coexpression Hotspots** representing
shared spatial pattern over many genes, describing
the multiscale spatial structure of the dataset

coex. hotspots find
structure at the scale it
occurs at, large or small

overlapping coex.
hotspots capture the
nested structure of the
hippocampus, identifying
all four major subareas.

**c**   Computation of Coexpression Hotspots

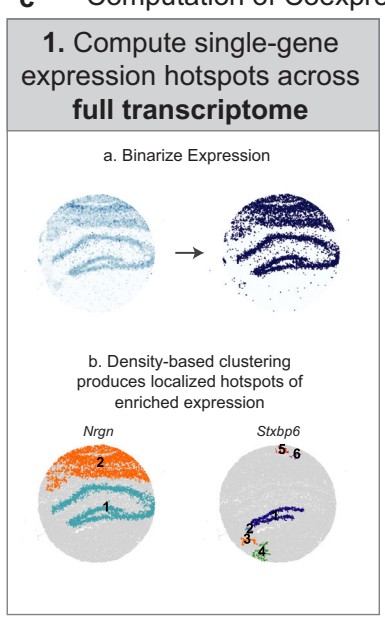

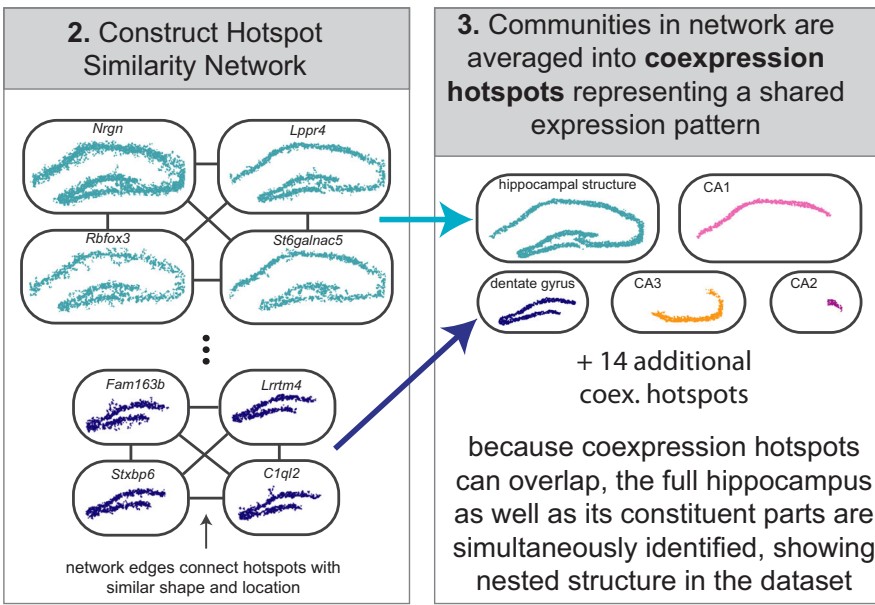

**d**   Downstream Analysis + Additional Features

spatially-aware
differential expression

decomposition of single-gene expression
into shared expression patterns

spatially localized
identification of cell-cell
interaction

3D hotspots + CCI
analysis

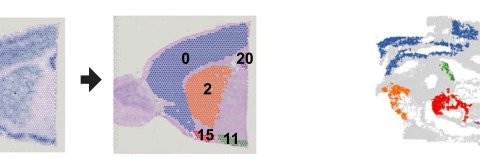

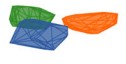

**Fig. 1 | NeST identifies nested, hierarchical structure in ST data through coexpression hotspot framework. a** Input ST dataset, up to full transcriptome coverage and potentially containing nested structure. **b** Output: coexpression hotspots represent contiguous areas enriched in some subset of genes. Compared to a segmentation, coexpression hotspots are scale-free and only identify structure where gene expression follows spatial organization. **c** Overview of coexpression hotspot computation. Note that NeST simultaneously searches for coexpression hotspots over any possible contiguous spatial subregion and any possible subset of genes across the full transcriptome. **d** NeST contains numerous additional features to facilitate further analysis of the ST data leveraging the coexpression hotspot framework.

## Four-layer nested structure in human breast cancer dataset

We next consider a Visium dataset of human breast cancer tissue and compute coexpression hotspots, observing both a high degree of overlap between coexpression hotspots and histology, as well as many overlapping hotspots indicating nested structure (Fig. 3a). Indeed, it is found that there are up to four layers of nested hotspots representing successively finer structure (Fig. 3b). Each successive layer expresses

additional cancer marker genes – considering top-1 markers, coexpression hotspot 0 (CH0) expresses DEGS2; CH2 expresses BRINP3; CH10 expresses SUSD3; and CH19 expresses LOXL2 (Supplementary Fig. 4a). This nested structure, however, is not captured by a segmentation (Fig. 3c). This dataset contains three nested structures: one with four layers, one with three, and one with two (Fig. 2d). Each of these represents areas of tumor tissue containing subregions

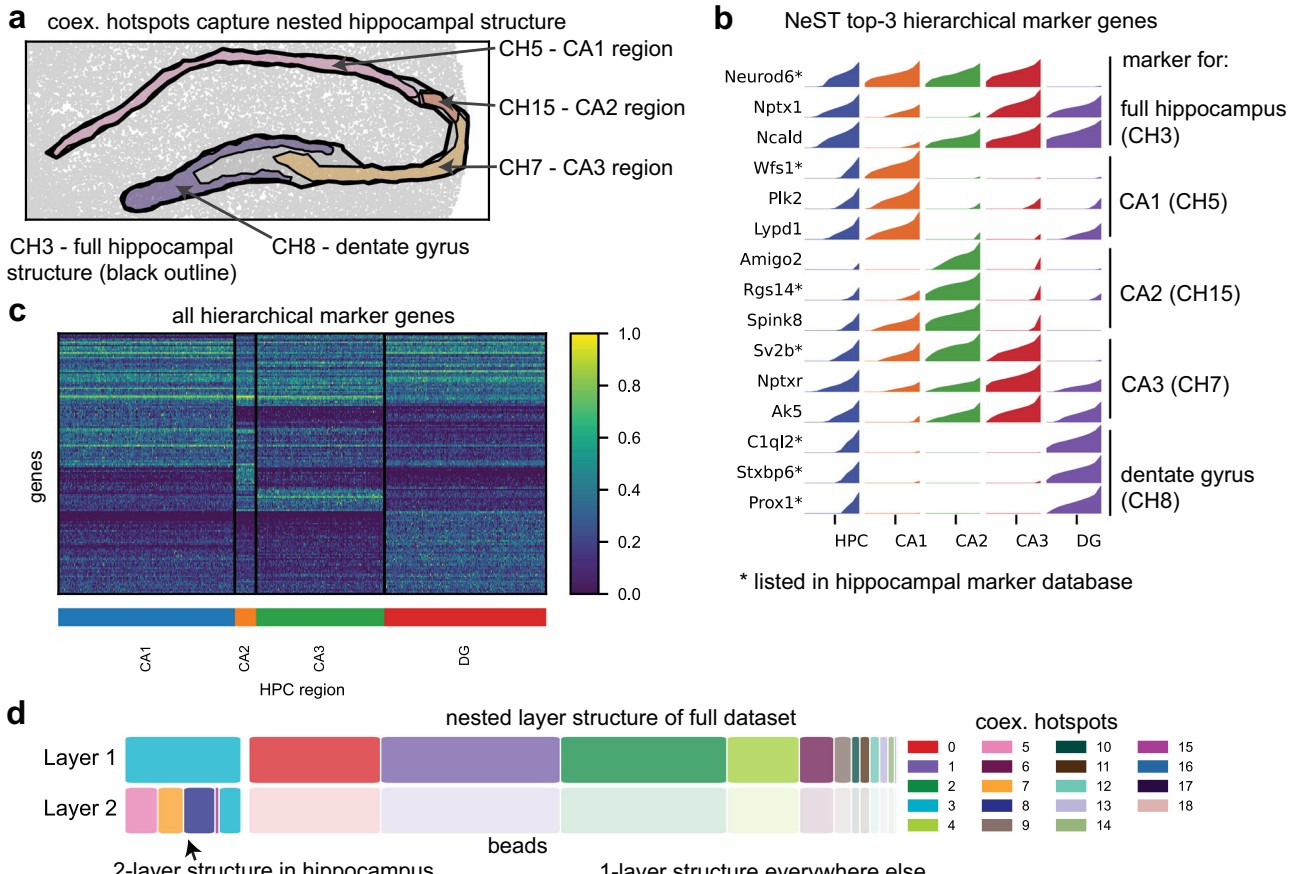

**a** coex. hotspots capture nested hippocampal structure

CH5 - CA1 region
CH15 - CA2 region
CH7 - CA3 region
CH3 - full hippocampal structure (black outline)
CH8 - dentate gyrus

**b** NeST top-3 hierarchical marker genes

marker for:
full hippocampus (CH3)
CA1 (CH5)
CA2 (CH15)
CA3 (CH7)
dentate gyrus (CH8)

* listed in hippocampal marker database

**c** all hierarchical marker genes

**d** nested layer structure of full dataset
coex. hotspots

2-layer structure in hippocampus
beads
1-layer structure everywhere else

**Fig. 2 | Coexpression hotspots capture nested hierarchical organization of hippocampus. a** Zoom-in on coexpression hotspots in the hippocampal formation, showing the five coex. hotspots that collectively represent the hierarchical organization. **b** Top-3 hierarchical marker genes (see Methods for definition) in each hippocampal hotspot, representing whole hippocampus, CA1, CA2, CA3, and dentate gyrus (DG), showing clear agreement with known marker genes[37]. **c** Heatmap of all identified hierarchical marker genes. Note the high overlap between CA2 and CA3 as well as very different expression in dentate gyrus. **d** Nested layer visualization, showing a second nested layer in hippocampus and single-layer structure elsewhere.

expressing additional genes, which cannot be represented under a segmentation framework. We also remark that the tree of nested groups produced by a hierarchical agglomerative clustering contains many clusters that are not spatially localized, as well as divisions of homogeneous regions into further subclusters (Supplementary Fig. 6).

Many regions of the tissue are not in any coexpression hotspot (see Fig. 3a) – this is because some regions do not contain groups of genes exhibiting similar spatial expression. To quantify this, we construct the spatial coherence score statistic, a normalized metric representing how many genes have spatially coherent expression patterns at each point in space (see Methods for details), showing a clear alignment with the known histology (Fig. 3e, see histology Supplementary Fig. 4b).

This tissue contains a tertiary lymphoid structure (TLS) which is of biological interest, and previous studies have used domain-specific knowledge to identify the presence of TLS[38-41]. However, by using the unique expression metric (Fig. 3f, see Methods), which identifies coexpression hotspots whose expressed genes are very different than the rest of the tissue, clearly highlights a single location – that of the TLS. This is because the TLS has a distinctive expression profile that is not present elsewhere in the sample, unlike the tumors which share similar expression across much of the tissue. Visualizations of marker genes for the corresponding coexpression hotspot confirm the presence of highly individual expression in this area including known TLS marker genes (Fig. 3g). Because NeST analyzes spatial coexpression across the entire genome, no prior knowledge of which genes are likely to be relevant is required.

## Benchmarking and validation

We compare NeST to two segmentation methods, HMRF[23] and SpaGCN[27], over a range of numbers of regions, on their ability to identify the top two layers of structure in the upper left tumor region, consisting of one outer region and three inner regions (Supplementary Fig. 5a, marker genes in Supplementary Fig. 5b). The structures can also be seen in the histology plot (Supplementary Fig. 4b). NeST produces a higher Jaccard score on the outer structure, indicating a better match with this structure, along with superior to equivalent performance on the inner structure depending on number of regions (Fig. 4a). When comparing the ratio between the size (number of spots contained) between the largest coexpression hotspot and the smallest, we see it is an order of magnitude higher than the ratio between largest and smallest regions for the HMRF and SpaGCN segmentation methods, over a wide range of numbers of regions (Fig. 4b). We also perform a similar comparison on the Slideseq dataset from Fig. 2, excluding SpaGCN due to performance, and similarly see a transition between detecting outer and inner structure at a setting of 25 regions (Supplementary Fig. 7). However, the HMRF method does not identify the CA2 region that NeST finds.

We perform comparison between the coexpression hotspots and those computed with random subsets of the total set of genes, and then for each coexpression hotspot in the full dataset, we compute the average Jaccard similarity with the best-match hotspot over 10 realizations of random subsets, both for subsets of 50% of the total genes (Fig. 4c) and 80% (Fig. 4d). We observe that almost all of the hotspots are not significantly affected by the removal of 20% of the genes, with a

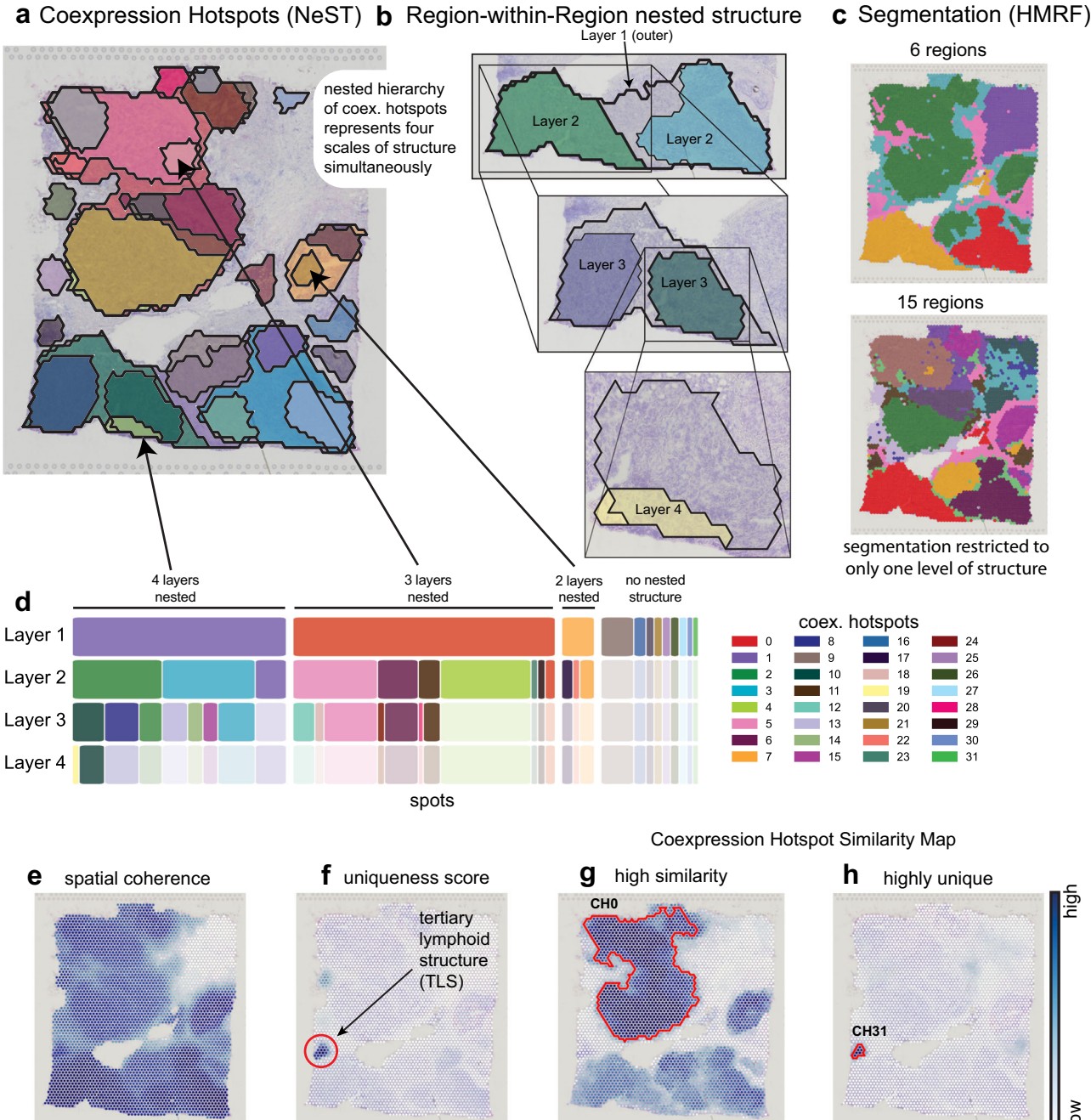

**Fig. 3 | NeST simultaneously identifies four layers of hierarchical organization in breast cancer tissue. a** Human breast cancer dataset showing all 32 identified coexpression hotspots. Note the large tumor regions containing three (top) and four (bottom) layers of structure. **b** Zoom in on the three additional layers of structure in the bottom tumor region. **c** Two example segmentations of the dataset computed with the HMRF method[23], which between them only capture two out of three layers in the top region and one out of four in the bottom region. In contrast, NeST captures all layers simultaneously. **d** Nested layer visualization for this

dataset, showing highly multi-layered nested hierarchical structure. **e** Spatial coherence score contrasts areas of tissue with spatially organized gene expression from those without, and shows high agreement with histology. **f** Uniqueness score shows areas with expression distinct from rest of dataset, highlighting tertiary lymphoid structure (TLS) with no prior knowledge. **g** CH0 shows high similarity to other tumor regions in the dataset. **h** In contrast, CH31, representing the TLS, has no similar areas elsewhere in the tissue, consistent with the uniqueness score shown in (**f**).

high average score in Fig. 4d. In the case that 50% of the genes are removed, all hippocampus hotspots except the very small CA2 are effectively preserved, but a number of hotspots towards the edge of the domain are largely lost. This provides a quantification of the sensitivity of each individual hotspot to the data.

In order to measure the specific effect that NeST tuning parameters have on the result, we constructed a synthetic five-layer hierarchical dataset by recursively dividing half of the domain into a new

region (see correct coexpression hotspots in Fig. 4e), with 2048 total genes of which a certain fraction contained spatial information, divided evenly over the regions. We varied each of the four main tuning parameters over a range, computed coexpression hotspots, and then compared those to the ground truth (Fig. 4f, see Methods for details). We observe that with as few as 64 out of 2048 spatial genes, NeST was able to effectively identify the regions in this dataset, whereas segmentation methods required a much higher number of spatial genes to

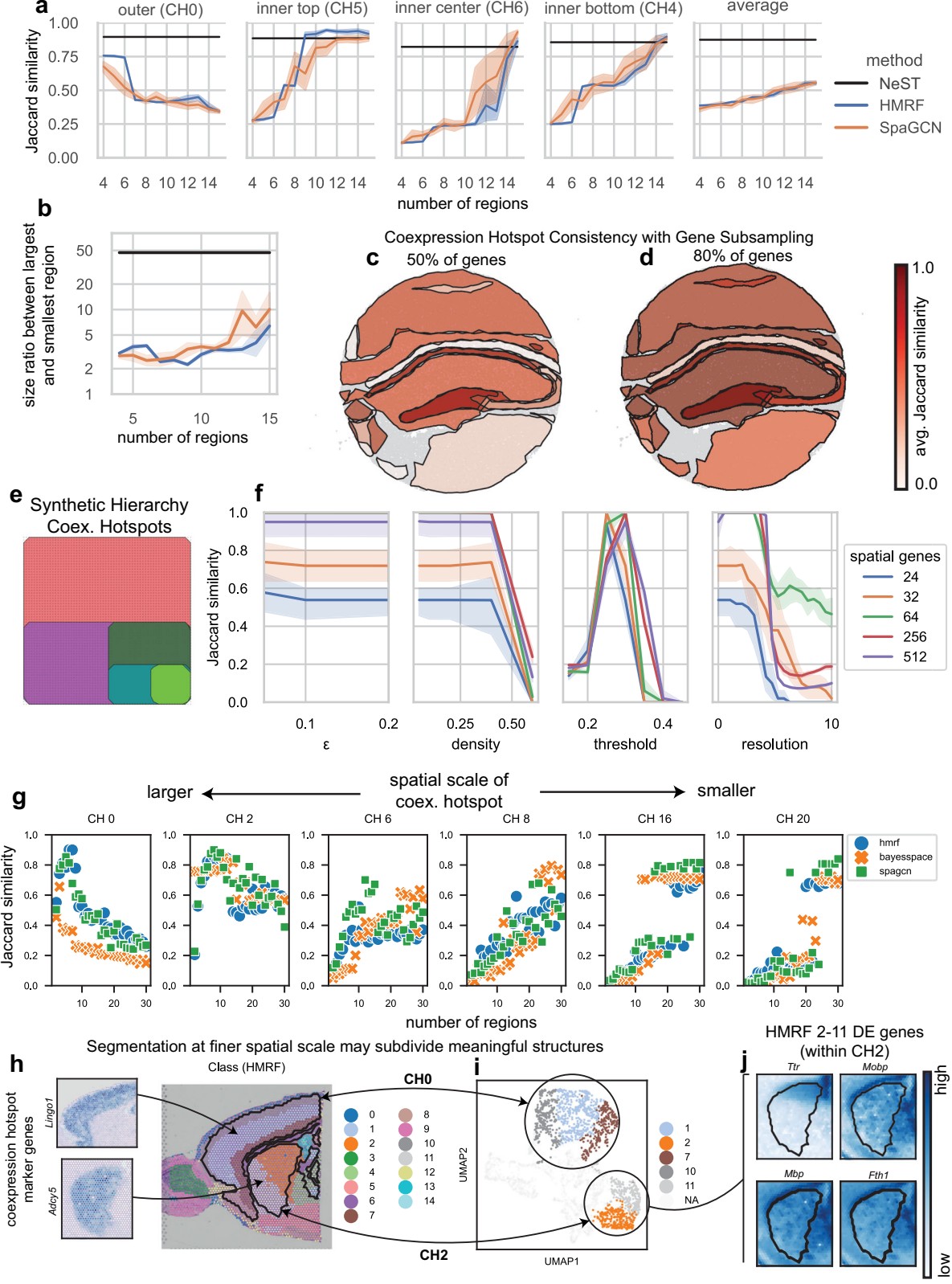

identify the structure (Supplementary Fig 8). Additionally, the critical values for the parameters at which the output stopped agreeing with the ground truth was largely insensitive to the signal-to-noise ratio, validating our approach of setting standard default values for these parameters.

We now consider a Visium dataset of the anterior mouse cortex, computing all coexpression hotspots with NeST as well as

segmentations using several methods: HMRF[23], BayesSpace[25], and SpaGCN[27], all of which are configured to identify a particular number of regions. By computing the overlap between segmentations and coexpression hotspots 0, 2, 6, 8, 16, and 20 (all of which are meaningful, substantiated by marker genes seen in Supplementary Fig. 9ab, Supplementary Fig. 10), the segmentation methods best identify large coexpression hotspots CH0 and CH2 when set to a coarse spatial scale

**Fig. 4 | Benchmarking and validation on real and synthetic data. a** Jaccard similarity between best-match region with each of the four areas representing the top two layers of structure in the upper-left tumor tissue. Line represents the mean, and shaded region represents the standard deviation across 10 randomly initialized realizations of each segmentation method. **b** The ratio of the size (number of spots) between the largest and smallest identified region, both for NeST (computed once, scale-free) and two segmentation methods across a variety of number of regions, measuring the range of spatial scales on which structure is identified. Line represents the mean, and shaded region represents the standard deviation across 10 randomly initialized realizations of each segmentation method. **c** Slideseq coexpression hotspots (using full dataset) colored by average Jaccard similarity over 10 realizations of coexpression hotspots computed using a random subset of 50% of genes. Darker color indicates the hotspot was more closely preserved with reduced genes. **d** The same as (**c**) but on subsets of 80% of the genes. **e** NeST coexpression hotspots on synthetic hierarchy dataset in a case where the hotspots accurately capture the full structure, consisting of 5 layers of nested structure. **f** Analysis of the

sensitivity of NeST coexpression hotspots to the four key parameters. Single-gene hotspots are computed from the set of cells that highly express a gene, and the fraction of cells within distance $\epsilon$ of that cell exceeds density. The hotspot similarity network is constructed with each hotspot as a node, and edges drawn between hotspots whose Jaccard similarity is greater than threshold. Leiden clustering is performed on the hotspot similarity network using resolution parameter resolution. Results are computed on the synthetic hierarchy dataset. Line represents mean, and shaded region represents 95% confidence interval across 10 independent synthetic datasets. **g** Comparison of NeST coexpression hotspots with the closest region from several segmentation methods (a score above approximately 0.75 suggests the regions match well). **h** Comparison of HMRF segmentation with coexpression hotspots, showing meaningful structures being subdivided. **i** UMAP visualization showing that the subdivisions of these regions are not just spatial but also reflected in expression space. **j** In the case of the central region (CH2), plotting top DE genes shows the region boundary is spurious, not reflected in actual genes. Source data are provided as a Source Data file.

(low number of regions), and small structures such as CH16 and CH20 when set to a fine spatial scale (large number of regions) (Fig. 4g), as expected. However, NeST is able to identify these structures on widely different spatial scales at the same time. Additionally, through hotspot decomposition, NeST is able to represent the overall expression pattern of genes in terms of coexpression hotspots, showing what parts of a gene's expression can be explained by spatial coexpression shared with other genes (Supplementary Fig. 9c).

We also illustrate an example of the HMRF segmentation where CH0 and CH2, have been subdivided (Fig. 4h). By visualizing the spot expression non-spatially as a UMAP plot, the divisions are also separated in expression space (Fig. 4i). Furthermore, in the case of CH2, which the HMRF segmentation divides into two regions, visualization of differentially expressed genes between the two regions shows that the expression patterns of the DE genes do not agree with the segmentation boundary (Fig. 4j). Thus searching for structure on a particular spatial scale may lead to large structures being unnecessarily subdivided. Conversely, at the scales at which the large structures are found, spatially small structures like CH16 and CH20 (coexpressing 120 and 994 genes respectively) are missed.

## NeST similarity maps identify related but spatially distinct structures

Given a coexpression hotspot representing a particular structure, NeST similarity maps show which areas of tissue also express a large fraction of those genes. Computing the similarity map for coexpression hotspot 8 (CH8), we see that the area of CH6 also lights up (Fig. 5a), indicating that the two coexpression hotspots have similar expression patterns and represent related structure. Inspection of the shared genes between CH6 and CH8 reveals marker genes such as *Olig1* as well as genes known to be involved in the myelination process[42] (Fig. 5b), suggesting that the CH6/CH8 structure is enriched in oligodendrocytes and may be involved in myelination processes.

Taking advantage of the fact that NeST represents these structures distinctly, we search for any possible differences between them. Performing DE analysis, we identify *Ccn2* as highly enriched in CH6 relative to CH8 (Fig. 5c). Visualizing the spatial gene expression, we observe that indeed *Ccn2* is expressed only in CH6, but not in CH8 (Fig. 5d). *Ccn2* has been tied to regulation of myelination[43,44], and so this suggests that *Ccn2* may modulate a difference in myelination behavior between the CH6 and CH8 regions. This observation also holds in a second dataset taken from another slice of the same anterior cortex (Supplementary Fig. 11a), validating its consistency. However, because the CH6 and CH8 regions are very similar, they are typically identified as the same region by segmentation methods, if they are identified at all (see Supplementary Fig. 11b). This highlights the importance of distinguishing between spatially disjoint regions with

extremely similar expression, as they may still have notable differences.

We also demonstrate the ability of NeST similarity maps to compare expression patterns across different samples. Taking a Visium dataset containing both a control and a disease (dextran sodium sulfate-induced colitis) sample of intestinal tissue, we test whether NeST can identify structures present in the disease condition but not the control condition. By taking the average similarity from the reference (disease) structure across the control-sample similarity map, we quantify how similar a particular coexpression hotspot is to a different sample. In the case that this average similarity is high, we identify shared structures across both datasets (Fig. 5e). In the case that it is low, we identify structures unique to the disease dataset (Fig. 5f). Above we introduced the uniqueness score, which identifies gene expression patterns localized to only one single area in a dataset. In contrast to this, the inter-sample comparison shown here identified patterns present, in any amount, in one dataset but not the other. This allows us to find differential patterns whether they are present only in one subsection of the sample or repeatedly across it.

We also show NeST analysis on a time sequence of developing mouse embryos[45], ranging from E9.5 to E16.5 in one-day intervals with one embryo per day (see Methods for details on datapoint selection). For coexpression hotspot 0 in the final datapoint (E16.5), representing the brain, we compute similarity maps over all seven previous datapoints, showing where in the earlier embryos similar genes were expressed (Fig. 5g). We show examples for CH1 through CH9 in Supplementary Fig. 12a–i, including other organs such as the liver (Supplementary Fig. 12a), heart (Supplementary Fig. 12b), and lung (Supplementary Fig. 12f), as well as examples of specific genes (Supplementary Fig. 13ab, 14, 15).

Finally, we show similarity maps for the Visium breast cancer dataset from Fig. 3. For a coexpression hotspot such as CH0, a top-level hotspot in the upper left hierarchical tumor structure, we see similarity across much of the tumor tissue in the dataset (Fig. 5h). In contrast, for the TLS, there is very low similarity anywhere else in the tissue (Fig. 5i), consistent with its identification as highly unique in Fig. 3h. We can also compute a dendrogram from the pairwise similarity values. Compared to the nested structure shown in Fig. 3 which shows the spatial relationships of hotspots, this shows the transcriptional similarity, and identifies that the tumor tissue can be split into two large groups (Fig. 5j, k). By combining spatially localized coexpression hotspots with similarity analysis, NeST simultaneously captures both spatial and transcriptional relationships between distinct structures in tissue.

## Spatial localization of cell–cell signaling within cell types
We next use NeST to show that spatial localization regions enriched in CCI differs significantly from cell type boundaries in a developing

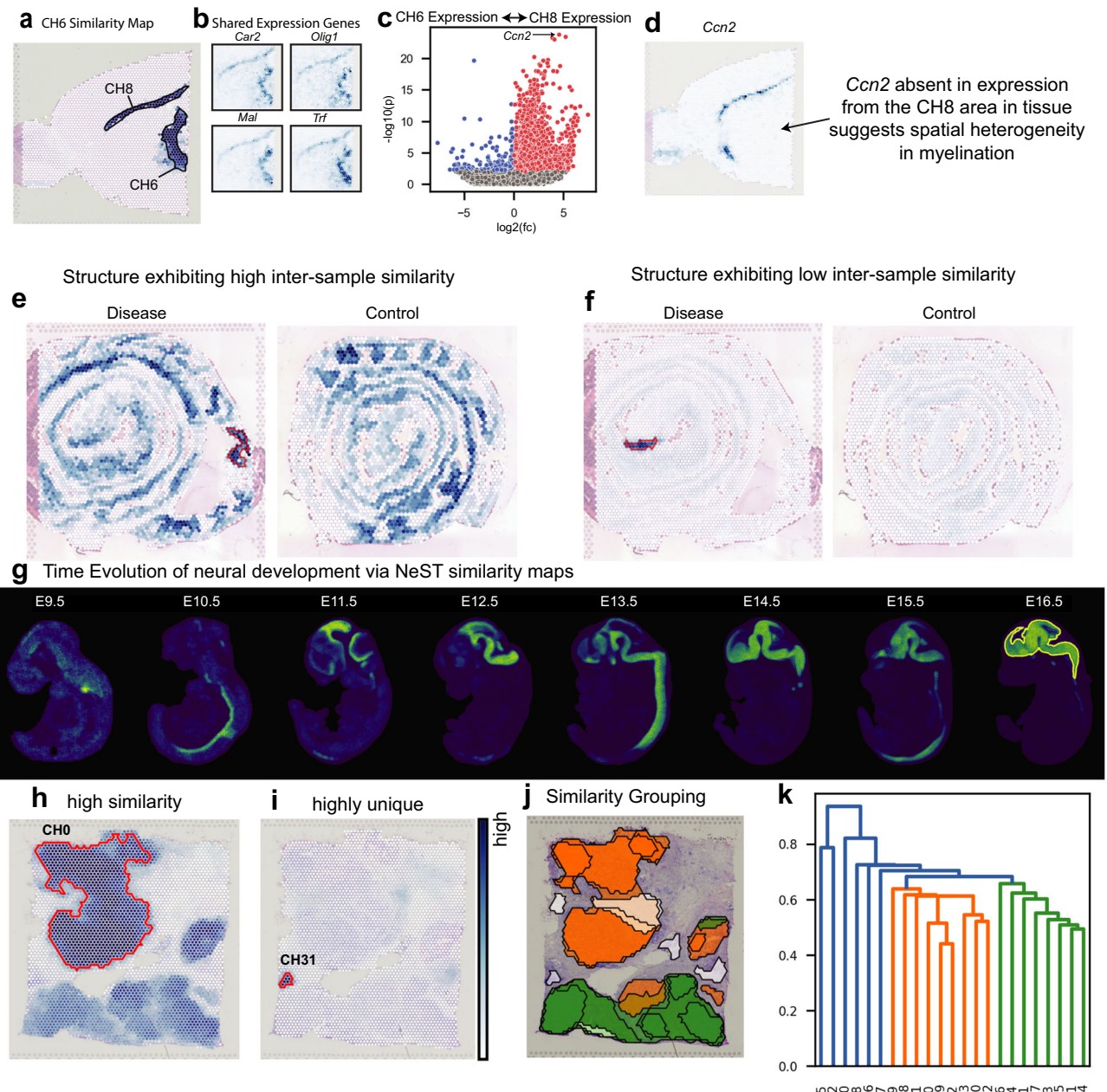

**Fig. 5 | NeST similarity maps identify repeated and unique structures.**
**a** Similarity map for CH6, an oligodendrocyte region, shows the presence of one other region, the CH8 region, with similar expression. **b** Many oligodendrocyte marker genes are enriched in specifically this region. **c** Differential expression between CH6 and CH8 shows Ccn2 is overexpressed in CH8 relative to CH6. Testing was performed by the two-sided Mann-Whitney *U* test combined with Benjamini-Hochberg FDR correction using $\alpha = 0.01$. Colored spots have a corrected *p*-value below 0.01. **d** Spatial visualization shows the expression of Ccn2 is entirely absent from CH6, in contrast to all other CH8 genes. **e** Similarity map for a coexpression hotspot from the disease sample (outlined in red), over both the disease and control sample, showing highly-similar structures repeated over both samples.

**f** Similarity map for a coexpression hotspot from the disease sample that does not have any similar structures, either elsewhere in the disease sample or in the control sample. **g** Similarity maps for CH0 of the E16.5 embryo (boundary highlighted) show the spatial patterns of expression of brain-related genes over the previous seven days of development. **h, i** Similarity maps for Visium breast cancer dataset show the distinction between coexpression hotspots formed by genes which are also expressed elsewhere in the tissue, such as CH0, and those formed by coexpression of genes that are entirely localized to that area, such as CH31, representing the tertiary lymphoid structure (TLS) present in the tissue sample (see Fig. 3ef). **j, k** Coexpression hotspots for grouped by a dendrogram computed from their pairwise similarity. Colors in (**i**) reflect groups from (**j**).

mouse embryo[9]. We remark that this dataset also contains nested structure visible in the brain region, with coexpression hotspots identifying both the full brain and the forebrain, midbrain, and hindbrain regions (Supplementary Fig. 16a, b). This structure is also not effectively identified by HMRF segmentation (Supplementary Fig. 16c). Just as coexpression hotspots freely identify gene coexpression where it occurs, not constrained to a single layer segmentation, CCI hotspots

identify CCI activity where it occurs and are not constrained to any preset partition, such as by cell type.

Taking as example the *Dll1-Notch1* ligand-receptor interaction, known to play a critical role in development[46], NeST identifies CCI hotspots based on a spatial diffusion model (Fig. 6a) to determine the level of ligand each cell is exposed to, and combining this with the receptor distribution (Fig. 6b) to determine an overall activity score

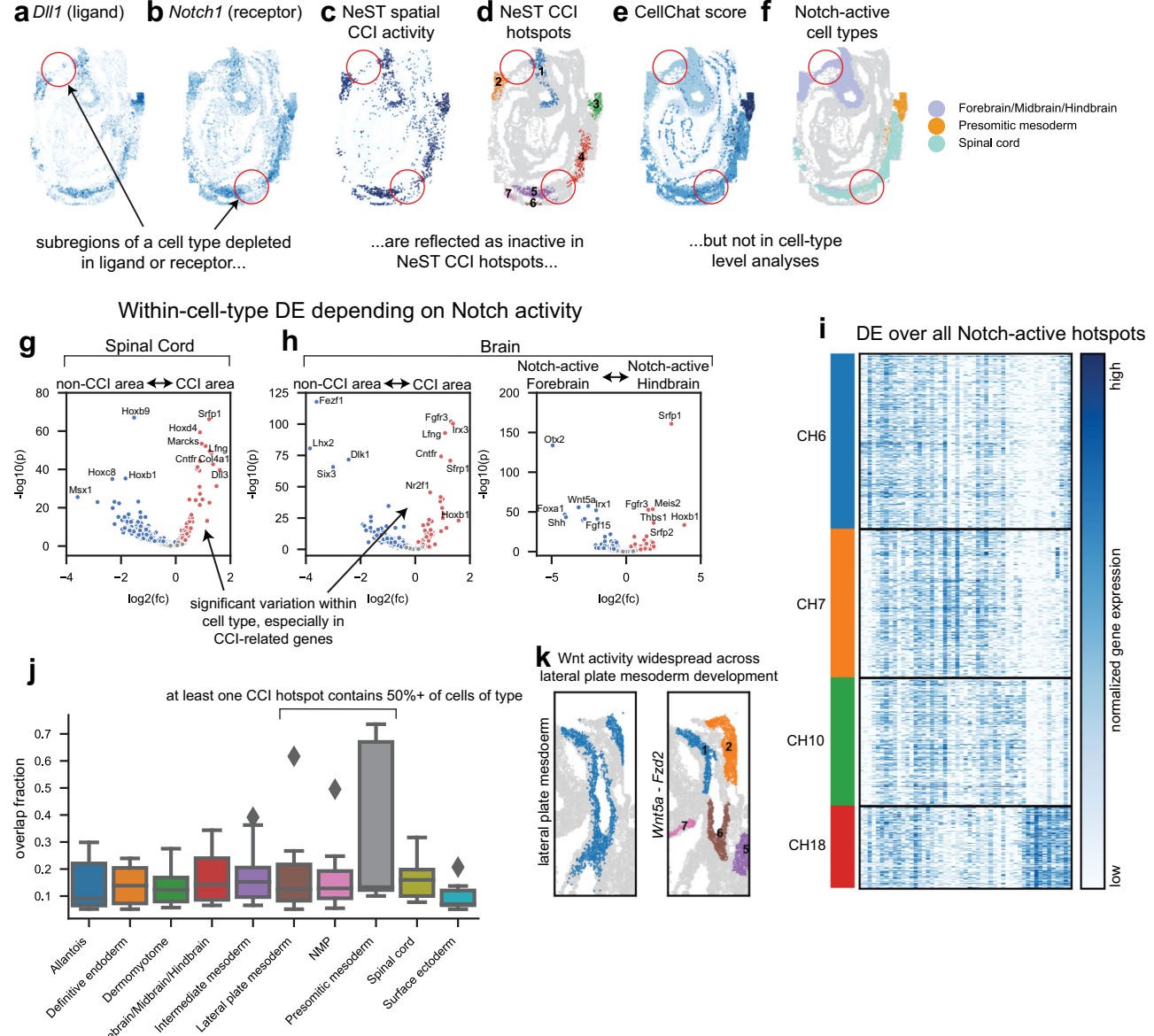

**Fig. 6 | NeST CCI hotspots localize regions of tissue active in CCI at single-cell, sub-cell type resolution. a** Distribution of normalized ligand expression for the Dll1-Notch1 interaction, showing spatially localized expression. **b** Distribution of normalized receptor expression (Notch1). **c** NeST spatial activity score, computed combining spatially diffused ligand expression (**a**) with receptor expression (**b**). **d** NeST hotspots for Dll1-Notch1, computed by applying the density-based clustering to the activity in **c** in the same way as single-gene hotspots are computed from gene expression. **e** Non-spatial computation of Dll1-Notch1 interaction using CellChat shows activity in many places where there is no expression of the ligand or receptor (see red circles). **f** The cell types exhibiting greatest activity according to CellChat[53] – see that the false positive regions are of the same cell type as cells that are active. **g** Differential expression within Spinal Cord cells, between those that are targets of Dll1-Notch1 interaction and those that are not. Testing was performed by the two-sided Mann-Whitney *U* test combined with Benjamini-Hochberg FDR correction using $\alpha = 0.01$. Colored spots have a corrected *p*-value below 0.01.

**h** Differential expression within brain cells, first based on Notch activity as in **g**, and then between the two active hotspots within the brain (corresponding to forebrain and hindbrain respectively). Testing was performed by the two-sided Mann-Whitney *U* test combined with Benjamini-Hochberg FDR correction using $\alpha = 0.01$. Colored spots have a corrected *p*-value below 0.01. **i** Heatmap of differentially expressed genes between all hotspots. **j** Distribution of overlap fraction with CCI hotspots for each cell type (over all 24 ligand-receptor interactions), showing that the overwhelming majority of hotspots do not agree closely with grouping by cell types. Box represents lower and upper quartile, center line represents median, whiskers extend to points within 1.5 times the inter-quartile range of the box, and outliers outside the whiskers are shown as diamonds. A total of 99 interaction hotspots over the 10 groups are included. All interaction hotspots are taken from the same dataset. **k** Comparison of Wnt5a-Fzd2 hotspots with lateral plate mesoderm cells, showing high overlap (see lateral plate mesoderm in **j**). Source data are provided as a Source Data file.

(Fig. 6c). Then, similarly to the single-gene case, density-based clustering extracts hotspots in which many active cells are clustered together (see Methods for details). Note that in spatial areas where either the ligand or receptor is absent (circled regions in Fig. 6a–f), NeST shows the lack of CCI activity, whereas computing CCI only

between or within cell types cannot make this sub-cell-type scale distinction.

The robustness of CCI hotspots is confirmed by the very high similarity in active target regions for the *Notch1-Dll1* and *Notch1-Dll3* interactions, as well as interactions from other pathways such as

*Fgf15-Fgfr1* (all interactions in Supplementary Fig. 17a). This is further reinforced by considering related functional genes such as *Lfng*[47,48], which is also observed to expressed in a similar spatial pattern (Supplementary Fig. 17b). This suggests biological significance of the four enriched regions, which appear as coexpression hotspots CH6, CH7, CH10, and CH18 (see Supplementary Fig. 9d for all coexpression hotspots). We refer to these four locations as Notch-enriched coexpression hotspots. Notch-enriched hotspots are found to have a higher coherence score than surrounding areas, indicating the presence of genes expressed specifically in these areas (Supplementary Fig. 17c). However, only CH18 lines up cleanly with cell type boundaries (Supplementary Fig. 17d) expressed in the presomitic mesoderm. CH7 and CH10 are subregions within the brain, corresponding to the hindbrain and part of the forebrain, and CH6 is a subregion within the spinal cord. In order to identify possible downstream effects or spatial correlations with Notch signaling, we compare cells that are targets of Notch signaling with other cells of the same cell type that are not targets of Notch signaling. Specifically, we first perform differential expression analysis between spinal cord cells that are contained within the spinal cord Notch-active coexpression hotspot CH6 and spinal cord cells not within CH6. We see enrichment of a number of genes in CH6 cells, including FGF pathway genes such as *Sfrp1* and *Fgfr3*, and a number of *Hox* genes, which do show expression specific to the CH6 area (Fig. 6g, Supplementary Fig. 17e). Similar analysis within the brain, comparing cells within brain Notch-active coexpression hotspots CH7 and CH10 to cells outside, shows significant DE between active and nonactive areas, with several CCI-related genes such as *Fgfr3* and *Lfng* highly enriched (Fig. 6h). Comparing CH7 cells with CH10 cells, the two Notch-active subregions within the brain, we see most notable differences in *Otx2*, known to be expressed in the forebrain[49,50], and *Sfrp1*, known to be expressed towards the hindbrain[51] (Fig. 6h). Visualizing DE across all four Notch-active coexpression hotspots simultaneously, CH18 is observed to have the most distinct expression pattern (Fig. 6i), consistent with its identity as the one Notch-active hotspot specific a unique cell type.

Under the hypothesis that CCI activity lines up with cell types, we expect the fraction of overlap between CCI hotspots and cells of a given type to be close to either 0 (not active) or 1 (active). However, only presomitic mesoderm, neural-mesodermal progenitors (NMP), and lateral plate mesoderm exhibit high CCI coverage, defined as the fraction of cells of that cell type that are contained within a CCI hotspot (Fig. 6j). The *Dll1-Notch1* CCI hotspots which exhibit highest overlap with NMP do not appear to form an NMP-specific structure (Supplementary Fig. 17f), but the *Wnt5a-Fzd2* interaction is widely expressed in a pattern specific to lateral plate mesoderm cells (Fig. 6k), which is consistent with prior study[52]. Overall, in most cases, CCI activity is heterogeneous within cell types, challenging the standard approach of computing CCI on a cell-type by cell-type basis.

### 3D NeST identifies Cck communication between layers and Tac signaling in behavior-associated regions in merFISH dataset

Finally, we use NeST to analyze three-dimensional spatial data using a merFISH dataset of the mouse cortex[6] containing approximately 74,000 cells over 12 distinct z-slices, each separated by a distance of 50 μm. The CCI inference proceeds similarly to above, however with a 3D diffusion model that allows for ligands to diffuse between different layers in the z-axis (Fig. 7a, see Methods for details). In this model, cells expressing the receptor may be activated by ligand expression by cells in other slices. We highlight a group of *Cck* expressing cells on a single layer surrounded by other cells expressing the *Cckbr* receptor, in which case a number of target cells on adjacent layers are identified through the 3D diffusion model (Fig. 7b) – inter-slice communication that could not have been detected through 2D analysis (Fig. 7c). Furthermore, the source cells are annotated as *Ambiguous*, and the target

cells as primarily *Excitatory* and *Inhibitory* (Fig. 7d, Supplementary Fig. 18a). All of these labels are spatially distributed through the entire region, so the spatial nature of this communication link would not be detectable through the typical cell-type-based analysis.

To better understand the three-dimensional structure of this dataset we compute three-dimensional regions (Fig. 7e, Supplementary Fig. 19, see Methods for details) as well as three-dimensional hotspots for cell–cell interactions (examples in Supplementary Fig. 18b–e). We illustrate the ability of NeST to find biologically meaningful functional regions by highlighting the case of the *Tac1-Tacr1* interaction in the top four slices (Fig. 7f). CCI hotspots allow us to distinguish between CCI active cells in different areas of the tissue, and so we zoom into the topmost slice for further comparison (Fig. 7g). When we perform non-spatial CCI analysis using CellChat[53], considering only the cell types, we observe that interaction is predicted even in areas of the tissue without ligand or receptor expression (Fig. 7h, c.f. ligand and receptor expression in Fig. 7i, j), underscoring the importance of correctly using spatial information when computing cell–cell interactions. We observe some genes, such as Avpr1a and Chat (Fig. 7k, l), appear to be enriched in the upper bilateral hotspots, and so we call these Chat+ hotspots. Comparing the two *Chat*+ hotspots to all other *Tac1-Tacr1* hotspots (in 3D), we can clearly see the difference in expression. In order to understand the role of these particular hotspots, we perform GO term analysis, finding enrichment of terms related to behavior, such as GO:0002118, *aggressive behavior*, and GO:0035176, *social behavior*, as well as many terms related to blood pressure due to the presence of *Avpr1a* (Fig. 7n). As *Avpr1a*, *Chat*, and *Oxtr* have been linked to behavior, we hypothesize that these cells represent a functional region in which interactions of *Tac* signaling and several other genes modulate behavior. When viewing the prevalence of different ligand-receptor interactions across z-slices, we see that there is significant heterogeneity between z-slices, with some ligand-receptor interactions enriched in lower, middle, or upper slices, further reinforcing the importance of capturing the full 3D behavior of cell-cell interactions (Fig. 7o).

## Discussion

Through its ability to identify nested, hierarchical, and multiscale structure in ST data, NeST represents an important next step in the method development of ST data analyses. NeST is released as a Python package and interfaces with the standard Anndata format[54] to allow easy application to new datasets. NeST is highly scalable; for example, computing coexpression hotspots on the Slideseq dataset with full transcriptome coverage and over 40,000 beads can be done in minutes on a standard laptop.

NeST allows for the identification of hierarchical structure as well as other spatially organized gene coexpression, and it contains a wide range of associated visualization tools in order to reveal the hierarchical structures in ST data, as well as compare spatial expression patterns within and between data samples. Beyond this, NeST leverages the unique nature of coexpression hotspots compared to traditional segmentations in order to allow for analyses such as spatial hierarchical marker genes, differential expression analysis between similar but spatially distinct structures, and functional analysis of single-cell resolution cell-cell interactions. NeST thus fulfills a previously unmet need in the analysis of spatial structure from ST data.

NeST is not a replacement for segmentation methods, but rather a new analysis offering additional tools to explore spatial gene expression patterns that have nested structure. The coexpression hotspot framework represents multiple layers of structure simultaneously, allowing analysis impossible with a segmentation; conversely, the segmentation approach captures the most dominant mode of spatial variation which can improve performance in some cases where structure is not clearly visible in any single gene.

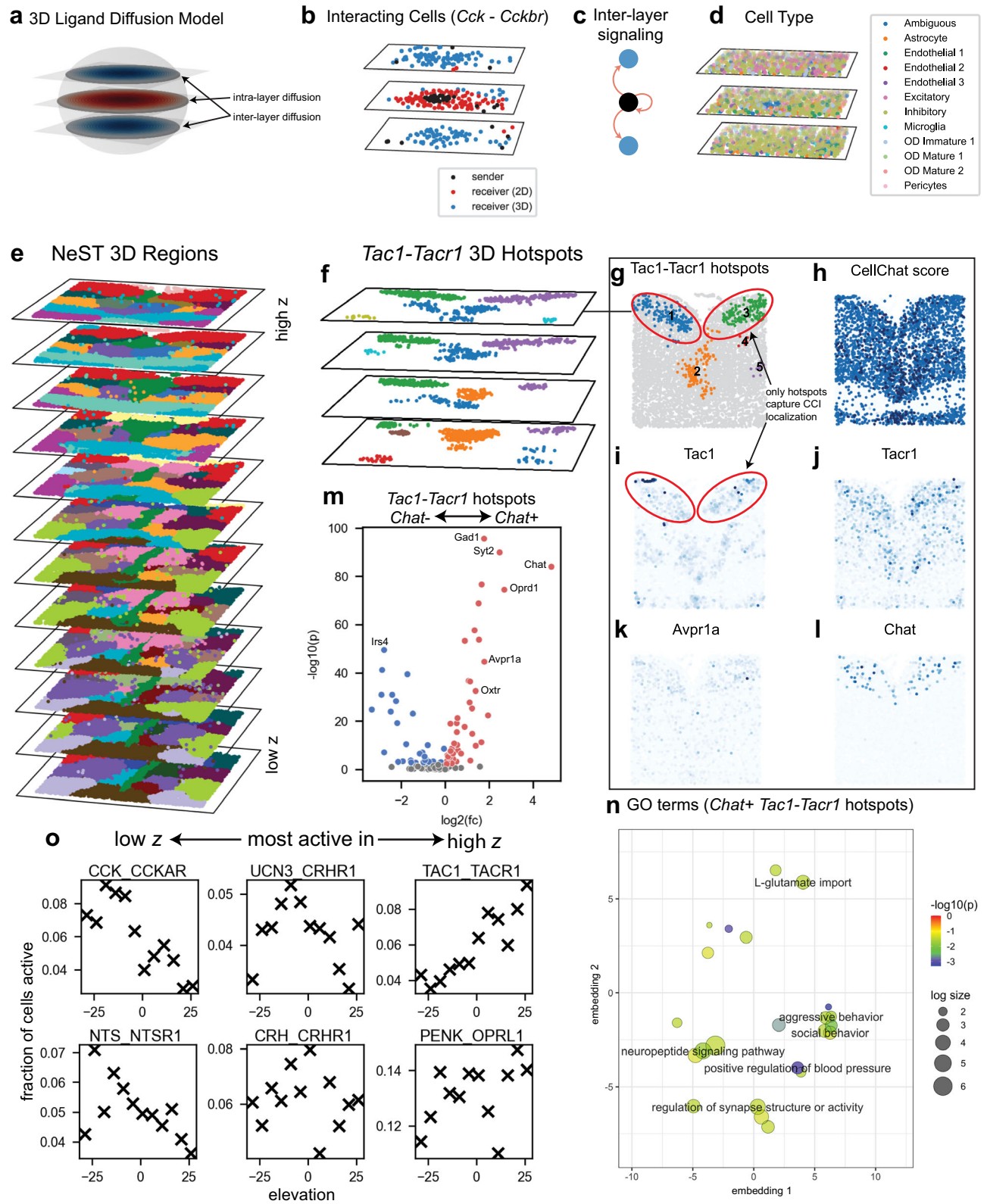

**a** 3D Ligand Diffusion Model

**b** Interacting Cells (*Cck - Cckbr*)

**c** Inter-layer signaling

**d** Cell Type

- Ambiguous
- Astrocyte
- Endothelial 1
- Endothelial 2
- Endothelial 3
- Excitatory
- Inhibitory
- Microglia
- OD Immature 1
- OD Mature 1
- OD Mature 2
- Pericytes

intra-layer diffusion
inter-layer diffusion

- sender
- receiver (2D)
- receiver (3D)

**e** NeST 3D Regions

**f** *Tac1-Tacr1* 3D Hotspots

**g** Tac1-Tacr1 hotspots

only hotspots capture CCI localization

**h** CellChat score

**i** Tac1

**j** Tacr1

**k** Avpr1a

**l** Chat

**m** *Tac1-Tacr1* hotspots
*Chat−* ⟷ *Chat+*

**n** GO terms (*Chat+ Tac1-Tacr1* hotspots)

**o** low *z* ⟷ most active in ⟶ high *z*

CCK_CCKAR   UCN3_CRHR1   TAC1_TACR1

NTS_NTSR1   CRH_CRHR1   PENK_OPRL1

fraction of cells active

elevation

One limitation of NeST in identifying coexpression hotspots is the initial step of computing single-gene hotspots. NeST relies on binarization for computation of single-gene hotspots, which could hide certain types of structures such as boundary regions with gradients in gene expression. This task bears a strong resemblance to the heavily-studied task of image segmentation, and more sophisticated processing such as incorporating convolutional neural network models could improve identification of single-gene hotspots. This could address a limitation in the DBscan-based clustering, which is not effectively able to find very thin layers whose width is less than the parameter ϵ. Additionally, computing single-gene hotspots while preserving continuous nature of gene expression could allow for our method for identifying coexpression to be extended to gradients, such as those found in developing embryos[55] or the brain[56]. An iterative method in

**Fig. 7 | NeST computes CCI hotspots fully in three dimensions revealing inter-layer communications and functional regions associated with social behavior in 3D mouse cortex. a** NeST 3D ligand diffusion model covers both intra- and inter-layer communication. **b** Cck ligand expression localized to one layer diffuses to receivers on adjacent layers over a subset of **c** NeST allows for identification of inter-layer communication that cannot be seen from single-layer analysis, where a sender cell in one layer (black) transmits a diffusive signal to adjacent layers (blue). **d** Cell type analysis in this region shows that CI does not align with cell type boundaries. **e** Regions identified in 3D by NeST. **f** Hotspots for Tac1-Tacr1 interaction over the top four layers. **g** Zoom-in on Tac1-Tacr1 hotspots in top layer. **h** Non-spatial cell-type-based analysis of Tac1-Tacr1 communication does not correctly identify spatial localization that is not based on cell type. Tac1 (**i**) and Tacr1 (**j**) expression agrees with the NeST hotspots. Upper bilateral hotspots are enriched in several genes such as Avpr1a (**k**) and Chat (**l**). **m** Differential expression analysis between Tac1-Tacr1 hotspots. Testing was performed by the two-sided Mann–Whitney U test combined with Benjamini-Hochberg FDR correction using $\alpha = 0.01$. Colored spots have a corrected p-value below 0.01. **n** GO term analysis for the Chat+ hotspots. Enrichment testing was performed by a two-sided Fisher exact test combined with Benjamini-Hochberg FDR correction using $\alpha = 0.05$. Terms with a corrected p-value below 0.05 were deemed significant. **o** 3D NeST CCI analysis identifies the z-axis spatial variation of ligand-receptor interactions. Source data are provided as a Source Data file.

which single-gene hotspots are refined based on information from tentatively computed coexpression hotspots may also increase performance through improved sharing of information across genes. Another avenue for improvement would be developing notions of significance, such as through statistical testing, that assess how unlikely a particular coexpression hotspot would arise through chance, which would further increase the ability of NeST to identify and highlight the most important spatial structures in a dataset. Furthermore, future work could seek to expand our notion of hierarchical marker genes and more extensively investigate the process of computing differential expression between overlapping groups of cells.

As spatial transcriptomic technologies continue to evolve, capturing more genes with greater efficiency and resolution, there is an ever-greater need for computational methods able to identify structure at multiple scales, filter out areas of increased interest, and substantiate the biological significance of identified spatial structure. The ability of NeST to identify multiscale, multilayer, explainable structure will open many new doors in the development of ST methodology and analysis of ST datasets.

## Methods

### Preprocessing
NeST is designed to be applied directly to full-transcriptome data and therefore no filtering of highly-variable genes, etc. is performed. Expression data is normalized and logarithmized before further analysis. For the Slideseq dataset, an additional spatial smoothing step is performed due to the high degree of spatial noise, in which the expression level at each bead is replaced by the average of the 20th and 80th quantiles of beads within a smoothing radius of 30 μm.

### Single-gene hotspots
We define a single-gene hotspot as a set of cells in a connected, localized subregion of space in which a particular gene is highly expressed. The first step is to binarize the data, which we perform using Otsu's algorithm[57], which divides the cells/spots into two groups such as to minimize the variance in expression within each group. Once the binarization has been performed, the locations of cells above the threshold are extracted, producing a set of two-dimensional points. We then apply DBscan density-based clustering[58] to this set, which first identifies core points, those for which at least *min_samples* other points exist with a radius ϵ. Then, all core points within radius ϵ are connected, and this produces the single-gene hotspots. The DBscan clustering is applied separately for each gene, but this process is not computationally intensive even without parallelization, being able to compute hotspots for all genes in a typical Visium dataset in under a minute. Optionally, after computing the hotspots, an $\alpha$-shape[59] boundary can be drawn enclosing the spots in the hotspot, and then the hotspot can be replaced with the set of all spots within the boundary. This means that the Jaccard similarity between hotspots (referenced below) corresponds to exactly the overlap in area between the hotspots. $\alpha$-shapes[59] are a generalization of convex hulls such that $\alpha = 0$ corresponds to the convex hull, and progressively larger values of $\alpha$ tighten the boundary, such that it becomes concave and more

closely surrounds the points. The shape becomes undefined for sufficiently large values of $\alpha$, and NeST uses a bisection algorithm to automatically select a large but valid value for $\alpha$. However, this comes at the cost of sparsity, and is not computationally tractable on single-cell resolution datasets.

Note that the computation of single-gene hotspots also serves as a filter for spatially-variable genes, as many genes whose expression does not follow a spatial pattern do not have sufficiently localized expression to have any single-gene hotspots identified, and therefore are filtered out from subsequent analysis.

### Coexpression hotspots
After computing all single-gene hotspots, we compute a similarity score between all possible pairs of hotspots. Here, we use the Jaccard similarity, which for two hotspots $H_i$ and $H_j$ is computed as

$$S_{ij} = \frac{\left| H_i \cap H_j \right|}{\left| H_i \cup H_j \right|} \tag{1}$$

The Jaccard similarity, being uniformly 0 for non-overlapping hotspots, leads to a sparse similarity matrix and can be efficiently computed even for very large numbers of hotspots, such as those arising from full-transcriptome gene hotspot computation. It can be rewritten as

$$S_{ij} = \frac{\left| H_i \cap H_j \right|}{\left| H_i \right| + \left| H_j \right| - \left| H_i \cap H_j \right|} \tag{2}$$

meaning the only pairwise relationship required is the (generally sparse) overlap matrix

$$O_{ij} = \left| H_i \cap H_j \right| \tag{3}$$

which represents the number of elements present in both hotspots. Instead of directly computing the pairwise overlap by iterating over every possible pair of hotspots, we iterate over every element (cell or spot in the ST dataset), identify the set of all hotspots containing that element, and then tally one overlap between every pair of hotspots in that set. As a result, non-overlapping hotspots do not consume any computation time. We combine this with a parallelized spatial chunking algorithm in order to maintain tractability even over full-transcriptome coverage, in which the dataset is divided into a rectangular grid (size does not affect results but we generally use $10 \times 10$) and the overlap counts are tallied separately in parallel for each grid square and then combined together.

The entries of the Jaccard similarity matrix over a user-defined threshold value are used as the weighted adjacency matrix defining the hotspot similarity network. Here, we use a threshold of 0.6 for Visium datasets and 0.3 for single-cell datasets (the lower threshold value due to the increased sparsity in single-cell datasets). Finally, we identify communities in the network using the Leiden algorithm[60]. Any

community with more than *min_genes* single-gene hotspots is used to create a coexpression hotspot. Given a group of single-gene hotspots, the corresponding coexpression hotspot is defined as the set of all spots/cells contained in over a certain fraction of constituent single-gene hotspots. Here, we use a value of 30%. The representation of the coexpression hotspot preserves reference to its constituent single-gene hotspots, so the set of genes being coexpressed can be utilized in downstream analysis.

## Hierarchical marker genes

In order to identify marker genes for nested hierarchical structures, NeST performs differential expression (DE) analysis using the two-sided Wilcoxon rank-sum test (also known as the Mann-Whitney U test) along with Benjamini-Hochberg FDR correction. For a given coexpression hotspot, those genes that are positively expressed at a significant level (here we use $p < 0.001$) over all parent, sibling, and child coexpression hotspots in the hierarchical structure plot are labeled as hierarchical marker genes for that coexpression hotspot. This can be further illustrated by using the NeST similarity map feature. Given a coexpression hotspot representing a particular structure, similarity maps show which areas of tissue also express a large fraction of those genes. encoding which hotspots are children of (i.e. contained within) other hotspots is computed by labeling any hotspot for which over 75% is contained within another hotspot as a child of that hotspot. In the case that there are multiple levels of nested structure, the parent of a hotspot is the smallest hotspot (i.e. the next level up) that contains at least 75% of its spots. Additionally, In the case that structure is not hierarchical, NeST can also compute marker genes over any user-provided set of coexpression hotspots using the same procedure.

## Coexpression hotspot decomposition

Once coexpression hotspots are computed, the expression of individual genes can be decomposed in terms of coexpression hotspots. NeST includes a procedure for identifying a subset of the identified coexpression hotspots that best matches a set of single-gene hotspots. For a particular gene, we define the match score as the number of spots in the single-gene hotspot that are also in the coexpression hotspot, minus the number of spots in the coexpression hotspot that are not in the single-gene hotspot. Letting $\{H_i\}_{i=1}^{N}$ be the set of $N$ single-gene hotspots for one particular gene and $H = \cup_{i=1}^{N} H_i$, then we the match score for coexpression hotspot $j$ is given as:

$$MS_j = \left| H \cap CH_j \right| - \left| \bar{H} \cap CH_j \right| \tag{4}$$

We take the hotspot $CH_k$ that maximizes this, $k = argmax_j MS_j$, add it to the decomposition, and then update H to reflect only the spots that are not covered by the decomposition:

$$H \leftarrow H - CH_k \tag{5}$$

where $-$ denotes set subtraction. This process is repeated, adding more coexpression hotspots to the decomposition, until no coexpression hotspot has a positive match score.

## Spatial coherence and unique expression score

We consider gene expression to be spatially coherent when many genes are expressed in the same subregion of tissue, with similar boundaries to the region of expression. We define a spatial coherence score to identify which areas of tissue exhibit the highest coherence, calculated by taking the subset of all single-gene hotspots that are a member of a coexpression hotspot and counting the number of such hotspots the spot or cell is contained in. Then, the score is normalized to range from 0 to 1 over all spots. In this way, areas in tissue with a large number of cells that are contained by many very similar hotspots have a higher spatial coherence score,

but since the score is computed in terms of not the coexpression hotspot but rather its constituent single gene hotspots, the spatial coherence score varies more smoothly than simply looking at coexpression hotspots. The unique expression score is computed by the same procedure, except the subset of single-gene hotspots is restricted to those hotspots that are a member of a coexpression hotspot, and no other hotspot of the same gene is a member of a different coexpression hotspot. This means we are identifying those genes that only have spatially coherent expression in one specific area of tissue, as opposed to genes that exhibit a repeated structure and are expressed around the tissue (see Fig. 3e, f).

## Cell–cell interaction hotspots

In this manuscript we perform CCI using the curated set of ligand-receptor interactions from Cellchat[53]. Specifically, we make use of the ligand gene symbol, receptor gene symbol, pathway name, and type annotation. We consider ligand-receptor pairs with annotations of "Secreted signaling" and "Cell-cell contact" in the database. Our method identifies all ligand/receptor pairs in the database for which both genes are present in the dataset and proceeds to perform analysis on those interactions.

Our method applies two ligand transport models: a diffusion-based model for secreted signaling interactions, and a neighbor-based model for cell-cell contact interactions. For each model, we construct a matrix $A_{ij}$ that represents the fraction of expressed ligand transported from cell $i$ to cell $j$. In the diffusion model, the ligand expression of a cell is distributed to all neighbors within a certain cutoff distance $\epsilon$, which we select to be 100 μm, expressed in the same spatial units as the data. The diffusion kernel is chosen to have a standard deviation of half the cutoff. The cell-cell contact matrix is constructed by taking the Delauney triangulation (where an edge between a pair of cells indicates that no other cell lies between them) and removing all edges over a certain threshold, which we take to be 20 μm. As a first-order correction for cell size, the transport matrices are normalized to have a row sum of 1. For each cell, we apply the ligand transfer model to spread its ligand expression over nearby cells, and then follow the procedure of CellChat[53], log-normalizing expression and then further normalizing to a maximum value of 1, and computing for each cell the product of the receptor expression and the transported ligand expression combined with a Hill function to determine the cell-level CCI activity. Letting $L_i^c$ and $R_j^c$ be the expression of ligand $i$ and receptor $j$ respectively for cell $c$, the activity is computed as:

$$\text{activity} = \frac{L_i^c R_j^c}{K_h + L_i^c R_j^c} \tag{6}$$

We generally take $K_h$ as 0.5, but as the Hill function is monotonic the output of the permutation tests described below are invariant to the choice of $K_h$.

In order to identify cells which exhibit a high level of activity, we perform permutation tests, computing $N_{perm}$ random permutations of the activity values. In each permutation, the gene expression vectors of each cell are shuffled across cells (applying the same permutation to each gene), while keeping spatial position the same. Then, the ligand transport model is applied to the shuffled expression and activation scores are computed for each permutation. We construct a distribution of null-hypothesis values by combining activation scores across all cells and all permutations. For a significance level $\alpha$, the significance cutoff for that interaction will be chosen as the $1 - \alpha$ quantile of the set of permuted activity scores. We then compute the binarized activation by testing the expression level of each cell against the cutoff for that interaction, and computation of hotspots then proceeds identically to gene expression hotspots as described above.

## Three-dimensional CCI analysis

For 3D datasets such as the merFISH dataset, the CCI can be run in 3D by providing an additional input representing the z coordinate value of each cell. The ligand diffusion model then uses 3D Euclidean distance, combined over all layers, instead of 2D Euclidean distance. Additionally, when performing permutation tests for significance, cells are only permuted within the same layer (cells with the same z-value). CCI hotspots are first computed individually for each layer, as described above, and then are matched across layers. To do this, we identify a nearest-cell-matching across each pair of adjacent layers using linear sum assignment with the cost set to the squared Euclidean distance. We then create a network, where each active cell (over all layers) is a node, and edges are drawn between each cell and its k-nearest neighbors ($k = 20$) in the same layer, as well as its matched neighbor in each adjacent layer. Intra-layer edges are weighted by the distance between cells i and j (in the same layer) as

$$w_{ij}^{\text{intra}} = e^{-0.04 d_{ij}} \tag{7}$$

where 0.04 is a constant chosen based on the distance units of this dataset. Inter-layer edges between cells i and j (in adjacent layers) are weighted by a factor $0.001\alpha^2$, where $\alpha$ is the number of k-nearest neighbors of cell i whose inter-layer matched neighbor is one of the k-nearest neighbors of j. This places greater weight on cells for which the inter-layer matching is spatially consistent. Then, communities are identified in this combined multi-layer network using the Leiden algorithm, using a RB vertex partition with resolution parameter $\gamma = 0.02$ for intra-layer edges and a CPM vertex with resolution parameter 0 for inter-layer edges, as is a standard for modularity optimization on multi-layer networks. The clusters output by the Leiden algorithm are taken as the combined multi-layer CCI hotspots. This matching process is performed separately for each interaction.

## Three-dimensional regions

To assist in visualization of three-dimensional structure in the merFISH dataset[6] along with three-dimensional CCI analysis, we identify regions using the Leiden multilayer network communication detection algorithm[60]. The network is constructed with both intra- and inter-layer edges, and intra-layer edges are derived from both spatial proximity and transcriptional similarity. For spatial proximity, the 20-nearest neighbor graph is used as the adjacency matrix, weighted by a factor of

$$A_{ij}^{\text{space}} = e^{\frac{d_{ij}}{0.04}} \tag{8}$$

where $d_{ij}$ is the distance between cells i and j and 0.04 is a weighting factor based on the scale of distances in the dataset, over all pairs i and j in the 20-nearest-neighbors graph. The expressional similarity matrix is given by

$$A_{ij}^{\text{expr}} = e^{-\frac{|z_i - z_j|}{2}} \tag{9}$$

where $z_i$ is the 8-dimensional PCA embedding of the expression of cell i. Finally, the full intra-layer adjacency matrix is given by

$$\mathbf{A}^{\text{inter}} = \alpha \mathbf{A}^{\text{space}} + (1 - \alpha) \mathbf{A}^{\text{expr}} \tag{10}$$

where $\alpha$ reflects the relative weighting between spatial and transcriptional similarity, here taken as $\alpha = 0.2$. The inter-layer edges are computing by performing a linear sum assignment between cells in every pair of adjacent layers under square-Euclidean cost. The communities labels are computed using the Leiden algorithm[60] using an RBConfigurationVertexPartition over intra-layer edges and a CPMVertexPartition with node_size = 0 over inter-layer edges.

## Parameter choice

Here we describe the parameters of NeST that may vary by dataset or analysis, as well as recommendations on how to set them when performing analysis.

## Alternative segmentation methods

The HMRF method was performed using a Python re-implementation of the HMRF algorithm included in NeST as nest.hmrf.HMRFSegmentationModel. For analysis, we used values $\beta = 1$ and a k-nearest-neighbors graph with $k = 6$. SpaGCN was run according to the tutorial values, running in the mode without histology image information, using k-means initialization and a maximum of 200 epochs although it reached convergence before the limit on tested datasets. A wrapper for SpaGCN replicating the analysis is available in nest as nest.methods.SpaGCN. BayesSpace was run using recommended values, with 2000 highly-variable genes and 16 principal components. A wrapper for BayesSpace using rpy2 replicating the analysis is available in nest as nest.methods.BayesSpace.

## CellChat

The database of ligand-receptor interactions from CellChat was used in determining what interactions to form CCI hotspots from. The CellChatDB.mouse database was loaded, and was filtered to use only secreted signaling and cell-cell contact interactions. This list was used as the database for NeST CCI hotspots.

The CellChat analysis of CCI was run following the procedure described in the original publication[53], with the communication probabilities computed using a truncated mean of 0.05. The exact procedure can be accessed through the nest package through the nest.methods.CellChat class. The CellChat score referenced in Fig. 7h was computed by taking the CellChat output, which is a communication weight for each pair of cell type, and then setting each individual cell to the sum of incoming weight for its cell type.

## Benchmarking

Due to the lack of ground truth annotation on the breast cancer dataset, the reference shown in Supplementary Fig. 3e was a manually curated combination of the models tested. Specifically, the HMRF and SpaGCN methods were each run twice, set for 4 and 15 regions, and then the output was separated into connected components to address the lack of localization in the segmentation methods, and the best-match label was manually identified for each of the four regions (outer and three inner). Then, combined with the best-match NeST coexpression hotspots, the reference was constructed as the set of spots assigned to that region by at least two of the three methods. We note that this is not meant to be taken directly as a ground truth, i.e. 1 being a perfect score, but rather a relative measure of how closely the output of the segmentation methods capture these structures compared to NeST.

Subsampling validation was performed by taking ten random subsets of the total set of genes in the datasets and independently computing NeST coexpression hotspots for each, and then these subsampled hotspots were compared to the original hotspots from the full dataset. For each of the original hotspots, and for each of the ten realizations, the Jaccard similarity to the best-match subsampled hotspot was computed. This was averaged over the ten realizations to produce the ultimate score. A score near one means that even with the subsampled set of genes, an identical coexpression hotspot is essentially always found, and a score near zero means that particular coexpression hotspot is no longer found with the reduced number of genes.

When benchmarking on the synthetical hierarchy data (see below for details on how the data is constructed), we take the default values for the four parameters from Table 1 and vary one of them away from the default at a time, and then compute coexpression hotspots. We

**Table 1 | List of NeST parameters**

| Name | Description | Suggestions to set |
|---|---|---|
| $\epsilon$ | DBscan clustering epsilon | For Visium or other multi-cell resolution spot array datasets, set to just above the distance between adjacent spots. For single-cell resolution data, a choice such that most cells have around 10–15 other cells within a radius $\epsilon$ is a reasonable default. If what appears to be one hotspot is split into multiple, increase $\epsilon$. If what appear to be distinct hotspots are merged together, decrease $\epsilon$. |
| local density | DBscan clustering min_samples divided by number of neighbors | Recommended default: 0.5 for multi-cell resolution, 0.2 for single-cell resolution. |
| Jaccard threshold | Minimum Jaccard similarity to form an edge in the hotspot similarity network | Recommended default: 0.6 if using hotspot closure (see below), 0.3 if not. Decrease if no edges are present in the hotspot similarity network. |
| resolution | Resolution parameter used in Leiden clustering of the hotspot similarity network | A default setting of 1.0 appears to work well in tested cases. |
| hotspot_min_size | Minimum number of spots/cells to form a single-gene hotspot | Set to the minimum number of spots/cells you would consider a meaningful structure in the dataset. |
| min_genes | Minimum number of genes to be coexpressed to form a coexpression hotspot | Try setting to a small value like 3, or increasing if the result contains undesirably many coexpression hotspots with very few genes. |
| Cutoff | Spots/cells in at least this fraction of single-gene hotspots are included in the coexpression hotspot | Default value: 0.3. Increase if coexpression hotspots appear smaller than their constituent single-gene hotspots, and decrease if they appear larger. |
| Spatial smoothing | Replace gene expression with average of 20th and 80th quantiles of nearby cells | Use on datasets with high (single-cell or near single-cell) spatial resolution but low capture efficiency such as Slideseq-v2. |
| Hotspot closure | Draw and fill in an $\alpha$-shape boundary around each single-gene hotspot | Use for multi-cell resolution data such as Visium. |

perform a one-to-one matching via linear sum assignment between the computed coexpression hotspot and the ground truth (overlapping) regions, attempting to maximize the Jaccard similarity of matched regions. In the case that NeST identifies too many or too few coexpression hotspots, those unmatched hotspots are given a score of zero. The scores are averaged over all matched pairs and unmatched hotspots to produce the Jaccard score shown on the y-axis of Fig. 4f.

Comparison between NeST and three segmentation methods: BayesSpace, HMRF, and SpaGCN, was performed on the mouse anterior cortex dataset by performing a series of segmentations with each method, varying the number of regions from 2 to 32, and then measuring the similarity between selected NeST coexpression hotspots that were observed to represent meaningful structure and the best-match region from each segmentation, as we vary the selected number of total regions in the segmentation. Here, we again remark that we are not assuming the NeST hotspot to be exactly the ground truth, but we have verified the significance of these structures by checking the expression of individual genes. A Jaccard similarity above approximately 0.75 should be considered a good match and therefore a success by the segmentation method in finding that structure.

### Synthetic data
In order to capture both hierarchical structure and multiple spatial scales, we consider a spatial structure consisting of a series of layers, each of which contains a new region covering half of the old. In other words, layer 1 covers the whole region, layer 2 covers half of the region, layer 3 covers half of layer 2, etc. Here we used a 5-layer dataset with a total of 2048 genes. Of these 2048 genes, a selected fraction were marked as spatial genes, assigned to one of the five layers, and then were expressed more heavily in that region than outside. Non-spatial genes have a spatially uniform expression distribution. Expression was modeled using a zero-inflated Poisson distribution, and the dataset was log-normalized.

### Stereo-seq dataset
The analysis shown on the Stereo-seq MOSTA dataset[45] consisted of one sample for each time point from E9.5 to E16.5, using the sample labeled E1S1 for each timepoint. $\epsilon$ values were scaled with the dataset as 0.02 times the length of the sample in the vertical direction. All samples used a density of 0.5, threshold of 0.3, and resolution of 1.0.

### Statistics and reproducibility
No statistical method was used to predetermine sample size. When selecting subsets of samples from collections of datasets, samples were always selected by numerically lowest label. Otherwise, no data were excluded from the analyses. NeST requires no filtering of highly variable genes and can be directly applied to full-transcriptome data. The experiments were not randomized. The Investigators were not blinded to allocation during experiments and outcome assessment.

### Reporting summary
Further information on research design is available in the Nature Portfolio Reporting Summary linked to this article.

### Data availability
All relevant data supporting the key findings of this study are available within the article and its Supplementary Information files. Visium 10x were accessed via SCANPY[54]. Other datasets were used through the Squidpy package[61]. Necessary code to load all datasets and use them with NeST is available as part of the NeST package. Raw forms of transcriptomic datasets are also available from the original authors. The Visium 10x datasets used in this study are available in 10x Genomics database at https://support.10xgenomics.com/spatial-gene-expression/datasets. The Slide-seqV2 dataset used in this study is available in the Single Cell Portal database at https://singlecell.broadinstitute.org/single_cell/study/SCP815/highly-sensitive-spatial-transcriptomics-at-near-cellular-resolution-with-slide-seqv2. The Seq-FISH dataset used in this study is available in the Spatial Mouse Atlas database at https://marionilab.cruk.cam.ac.uk/SpatialMouseAtlas/. The MERFISH dataset used in this study is available in Dryad at https://doi.org/10.5061/dryad.8t8s248. The intestine colitis Visium dataset used in this study is available in the GEO database under accession code GSE169749. The Stereo-seq data of mouse embryo development is available in the CNGB database under accession code CNP0001543 or at https://db.cngb.org/stomics/mosta/download/. The CellChat database of ligand-receptor interactions used in this study is part of the CellChat R library available at https://github.com/sqjin/CellChat. Source data are provided with this paper.

### Code availability
NeST is available as a Python package and can be accessed at https://github.com/bwalker1/NeST[62].

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

## Acknowledgements

The project was partly supported by the National Science Foundation grants DMS176372 and CBET2134916, the National Institutes of Health grants U01AR073159, R01DE030565, and R01AR079150, and a Simons Foundation Grant (594598), awarded to Q.N.

## Author contributions

B.W. and Q.N. conceived the project; B.W. implemented the algorithm and code and conducted data analysis. All the authors wrote and approved the manuscript; Q.N. supervised the research.

## Competing interests

The authors declare no competing interests.
