## [Peer Review File · Nature Communications]

Nested Hierarchical Structure in Spatial Transcriptomic Data with NeSTEditorial Note: This manuscript has been previously reviewed at another journal that is not operating a transparent peer review scheme. This document only contains reviewer comments and rebuttal letters for versions considered at Nature Communications.

Reviewer #1 (Remarks to the Author):

We appreciate the effort of the authors in addressing our technical concerns. These have been largely properly addressed in our view, and we think that the work is statistically sound.

However

1. We remain unconvinced of the biological relevance of the approach, which can not be validated. To us it seems it would be useful just as a visualization strategy.
2. We are not convinced about the approach for the multi-sample analysis, and if it would be helpful in the analysis of large cohorts.
3. Their benchmark with other methods is based on finding regions that others do not, but these are not real 'true positives' or 'meaningful clusters', as the ground truth is unknown.

Reviewer #2 (Remarks to the Author):

Summary

The authors have made good progress toward addressing most of my initial comments. However, there are a few remaining issues.

Major

- * In two places the authors say that agglomerative hierarchical clustering does not produce "meaningful" results because it can only merge two clusters at a time. I do not find this argument convincing. While it's true that the most common agglomerative clustering algorithms merge pairs of clusters, there is no reason in principle that 3-way or 4-way merges could not be considered. Would that then lead to "meaningful" results? I doubt it. The authors need to better define "meaningful" (or preferably use a more precise term). One obvious difference between NeST and hierarchical clustering is that the latter ignores spatial information.
- * "concave boundary"- I still don't know what this is. Can the authors please describe with math or a diagram (maybe in the supplement)? I asked in my previous comments whether they really meant a convex hull, but they didn't directly answer the question.
- * results section "NeST similarity maps identify" paragraph 4,5: when should we use similarity maps versus uniqueness scores?
- * Methods single gene hotspots- "all core points within radius ϵ are connected.". Is it really only the core points or is it actually all of the points? The way it's written it seems like non-core points can never be connected.
- * Table 1- "Visium data"- I think what you mean is, any data where a single spot has multiple cells. "Single-cell"- does this also include protocols with resolution of less than single cell (I think merfish can do this?). Also, under "spatial smoothing"- you mention "Slideseq", is this V1 or V2?

Minor

- * Figure 3- there is no caption for panel (g).
- * Figure 4- explain what ϵ means, actually it might be good to just give this an understandable name and use it throughout the text and in Table 1 instead of as a Greek letter.
- * Figure 5a- show the outline of CH6 and CH8 with labels if possible.
- * "single the score" in methods- I think you mean "since the score"
- * "In each permutation, the gene expression vectors of each cell are shuffled..."- are the genes shuffled jointly or is each gene shuffled independently?
- * Methods benchmarking section, last sentence "therefore a total success"- remove the word "total".
- * The software installation instructions are improving but I still got an error when trying to import nest:
`ModuleNotFoundError: No module named 'colorcet'` perhaps this needs to be added to the conda environment file.

Responses to Reviewer 1

We appreciate the effort of the authors in addressing our technical concerns. These have been largely properly addressed in our view, and we think that the work is statistically sound.

We thank the reviewer again for their help and are glad to hear that the reviewer agrees with the statistical soundness of our method with the added benchmarking. In this revision, we have made numerous changes in order to show the ways in which NeST analysis can be validated and substantiated using biological knowledge, and we have added another example of using NeST for comparison of multiple samples, this time on a dataset capturing the time-evolution of developing mouse embryos. We hope that with this revision the significance of our method will be clear. Responses to specific points follow below.

However

1. We remain unconvinced of the biological relevance of the approach, which can not be validated. To us it seems it would be useful just as a visualization strategy.

We appreciate the reviewer's point on the important question of establishing biological relevance and validation of the approach. First, we recognize the importance of verifying that coexpression hotspots do reflect underlying structure in the gene expression, and in this revision, we have newly added additional supplemental figures that show further examples of such spatially expressed genes (SI Fig. 2, 5b, 10, 14, 15).

The significance of identifying spatial regions of biological meanings in ST data is well established in the literature, and NeST expands on this with a novel approach that allows for identification of *hierarchical structure* as well as numerous other benefits as shown in the manuscript. In terms of validating and justifying the biological relevance of coexpression hotspots,

- In Fig. 2 (hippocampal structure), Fig. 3 and SI Fig. 4 (breast cancer), and SI Fig. 16a (mouse embryo brain), we show the hierarchical structure identified by NeST agrees well with known biological structures, as shown with confirming agreement with known marker genes.
- We show the histology is in good agreement with coexpression hotspots (SI Fig. 4bc).
- And we show agreement between CCI hotspots and known related functional genes (SI Fig. 10).

We feel this serves as a good justification for the biological relevance of NeST analysis.

While the various visualization tools offered by NeST are a key part of its usefulness, the additional analysis possible with hotspots also show how NeST can be used beyond just as a visualization tool, such as:

- The ability to identify the presence of multiple layers of hierarchical structure.
- The ability to compute hierarchical marker genes (Fig. 2bc, SI Fig. 4).
- Analyses of differential expression to identify fine differences between related hotspots such as shown in Fig. 5c, Fig. 6gh, and Fig. 7m.
- Identification of spatial cell-cell communication at single-cell resolution and in three dimensions (Fig. 6, Fig. 7).

Overall, we feel that NeST serves a valuable purpose in both visualization and analysis of ST data. We have added the following text to the discussion in order to address this point:

“NeST allows for the identification of hierarchical structure as well as other spatially organized gene coexpression, and it contains a wide range of associated visualization tools in order to reveal the hierarchical structures in ST data, as well as compare spatial expression patterns within and between data samples. Beyond this, NeST leverages the unique nature of coexpression hotspots compared to traditional segmentations in order to allow for analyses such as spatial hierarchical marker genes, differential expression analysis between similar but spatially distinct structures, and functional analysis of single-cell resolution cell-cell interactions. NeST thus fulfills a previously unmet need in the analysis of spatial structure from ST data.”

2. We are not convinced about the approach for the multi-sample analysis, and if it would be helpful in the analysis of large cohorts.

We thank the reviewer for their comment. In order to further demonstrate the usage of similarity maps across multiple samples, In the revision, we have added a new analysis showing a Stereo-seq dataset of mouse embryo development. Specifically, we show simultaneous analysis of 8 embryos ranging from E9.5 to E16.5 in one-day intervals. By selecting a NeST coexpression hotspot in the final embryo, we compare to each previous day, thus tracing out where the genes that were enriched in a particular location at the final timepoint had been expressed in the past. These are shown in Fig. 5g and SI Fig. 12, and the dataset is further shown in SI Fig. 13-15. In the results section, we have added the following text:

“We also show NeST analysis on a time sequence of developing mouse embryos⁴⁵, ranging from E9.5 to E16.5 in one-day intervals with one embryo per day (see Methods for details on datapoint selection). For coexpression hotspot 0 in the final datapoint (E16.5), representing the brain, we compute similarity maps over all seven previous datapoints, showing where in the earlier embryos similar genes were expressed (Fig. 5g). We show examples for CH1 through CH11 in SI Fig. 12, including other organs such as the liver (SI Fig. 12a), heart (SI Fig. 12b), and lung (SI Fig. 12f), as well as examples of specific genes (SI Fig. 13, 14, 15).”

We hope that these examples will provide readers with an interest in multi-sample analysis to determine if NeST might be useful for their purposes.

3. Their benchmark with other methods is based on finding regions that others do not, but these are not real 'true positives' or 'meaningful clusters', as the ground truth is unknown.

We thank the reviewer for highlighting the importance of establishing the significance of the basis used for the comparison between methods. Regarding the comparison on the breast cancer dataset shown in Fig. 4ab, we have added SI Fig. 5 in order to show marker genes for each of the regions. We also note that all of these regions are clearly

visible directly in the histology image (SI Fig. 4b). In the results section, we now introduce these regions as follows:

“We compare NeST to two segmentation methods, HMRF and SpaGCN, over a range of numbers of regions, on their ability to identify the top two layers of structure in the upper left tumor region, consisting of one outer region and three inner regions (SI Fig. 5a, marker genes in SI Fig. 5b). The structures can also be seen in the histology plot (SI Fig. 4b).”

For the comparison on the cortex dataset shown in Fig. 4g, in addition to the existing SI Fig 7b showing the constituent single-gene hotspots, we have also added visualization directly of each of the relevant genes in order to ensure that the selected hotspots can be directly viewed in terms of gene expression, shown in the new SI Fig. 10.

Given that all of the structures used in comparison can be clearly visualized in terms of gene expression pattern consistent with histology results, we naturally consider them as “meaningful structures”.

Responses to Reviewer 2

Summary

The authors have made good progress toward addressing most of my initial comments. However, there are a few remaining issues.

We thank the reviewer for considering our previous revision. We recognize that there were several important points raised in the previous review that were not clearly or satisfactorily addressed, and we have sought to be much more specific in the requested explanations. Responses to specific points follow below:

Major

* In two places the authors say that agglomerative hierarchical clustering does not produce "meaningful" results because it can only merge two clusters at a time. I do not find this argument convincing. While it's true that the most common agglomerative clustering algorithms merge pairs of clusters, there is no reason in principle that 3-way or 4-way merges could not be considered. Would that then lead to "meaningful" results? I doubt it. The authors need to better define "meaningful" (or preferably use a more precise term). One obvious difference between NeST and hierarchical clustering is that the latter ignores spatial information.

We apologize for not clearly addressing this important point in the prior revision and agree that the argument as presented did not clearly emphasize the difference between the hierarchical structure identified by NeST and that by a typical agglomerative clustering algorithm.

In our study, a "meaningful" hierarchical structure is one where subclusters are only joined together if they both share transcriptional similarity, and are also nearby, and conversely clusters are not divided into subclusters unless there is a transcriptional distinction. Simply applying a non-spatial agglomerative clustering algorithm, as shown in SI Fig. 6, produces a hierarchy containing in total thousands and thousands of clusters, virtually all of which violate this property (the "not meaningful" as we used it).

It is of course conceivable that someone could attempt to develop a method that takes a hierarchy from such a method and extracts out only the small subset of "good" clusters. But to our knowledge, such a method does not currently exist in the ST literature. Instead, one could imagine using the spatial information directly to make a hierarchy that is both spatially and transcriptionally consistent from the beginning, requiring no such pruning – and this is precisely what we have made with NeST.

In order to further show that the coexpression hotspots from NeST represent spatially consistent regions consistent with the above, we have newly added additional supplemental figures that show further examples of such spatially expressed genes (SI Fig. 2, 5b, 10, 14, 15).

In addition to these new results in the revision, we have adjusted the two locations in the manuscript addressing this point to read:

"In contrast, the hierarchical structure produced with standard agglomerative hierarchical clustering methods, created by repeatedly merging smaller clusters into larger, contains

clusters that do not represent spatially localized regions, or fail to combine spatially adjacent and transcriptionally similar cells.” (Introduction)

“We also remark that the tree of nested groups produced by a hierarchical agglomerative clustering contains many clusters that are not spatially localized, as well as divisions of homogeneous regions into further subclusters (SI Fig. 4).” (Results)

* "concave boundary"- I still don't know what this is. Can the authors please describe with math or a diagram (maybe in the supplement)? I asked in my previous comments whether they really meant a convex hull, but they didn't directly answer the question.

We apologize for the lack of clarity and direct response to the previous point. The method used to draw these boundaries creates what is typically referred to as an α -shape, and we have made correction to use the proper terminology. We have added a reference to the formal mathematical definition of α -shapes, as well as a short intuitive explanation, which reads:

“ α -shapes are a generalization of convex hulls such that $\alpha=0$ corresponds to the convex hull, and progressively larger values of α tighten the boundary, such that it becomes concave and more closely surrounds the points. The shape becomes undefined for sufficiently large values of α , and NeST uses a bisection algorithm to automatically select a large but valid value for α .”

* results section "NeST similarity maps identify" paragraph 4,5: when should we use similarity maps versus uniqueness scores?

We thank the reviewer for their question on these different types of analysis. In short, the uniqueness score addresses the question “What gene expression patterns are localized to only one single area in this dataset?” whereas the inter-sample comparisons in this section answer the question “What gene expression patterns are present in this dataset, but not the other?”. Intuitively, if a structure was present in lots of different locations in the disease dataset, but nowhere in the control dataset, that would be highly interesting, even though it would not score highly on the uniqueness score since it is not localized to a single area. In order to make this distinction clearer, we have added the following text to this section:

“Above we introduced the uniqueness score, which identifies gene expression patterns localized to only one single area in a dataset. In contrast to this, the inter-sample comparison shown here identified patterns present, in any amount, in one dataset but not the other. This allows us to find differential patterns whether they are present only in one subsection of the sample or repeatedly across it.”

* Methods single gene hotspots- "all core points within radius ϵ are connected.". Is it really only the core points or is it actually all of the points? The way it's written it seems like non-core points can never be connected.

Non-core points are indeed never included in the output single-gene hotspots. This is intentional, and has the desired effect of filtering out individual cells that may highly

express a gene but are not within a region containing a number of other cells also expressing the gene. It is true that in more traditional DBscan clustering, those non-core points that are within a distance ϵ of at least one core point are included in the clusters. However, we found that this often led to the inclusion of extraneous cells right outside the border of a meaningful spatial structure and so decreased the clarity of the result, and so in our implementation we chose to exclude them. By passing the optional argument ``core_only=False`` to the NeST function ``compute_gene_hotspots``, a user could compute hotspots with non-core points added to the periphery if desired.

* Table 1- "Visium data"- I think what you mean is, any data where a single spot has multiple cells. "Single-cell"- does this also include protocols with resolution of less than single cell (I think merfish can do this?). Also, under "spatial smoothing"- you mention "Slideseq", is this V1 or V2?

We thank the reviewer for their suggestions on improving terminology for different types of data. Indeed, for Visium we were referring to multi-cell resolution spot datasets (where in this manuscript we used only Visium type), and we have adjusted Table 1 to describe datasets using *"single-cell resolution"* and *"multi-cell resolution"*.

Regarding sub-cellular resolution like MERFISH, these data are generally combined into whole cell resolution through a cell segmenting process. There is a fundamental difference between the biological meaning of spatial variation in gene expression between different cells (what NeST analyzes), and spatial variation in the location of transcripts within a particular cell, so we do not address here the analysis of MERFISH data without cell segmentation.

We have only tried NeST on Slideseq-v2 data, not v1, and have updated the example we gave in the table to specify *"Slideseq-v2"*, but the suggestion would be the same either way.

Minor

* Figure 3- there is no caption for panel (g).

We apologize for the omission and have added an appropriate caption for panel (g).

* Figure 4- explain what ϵ means, actually it might be good to just give this an understandable name and use it throughout the text and in Table 1 instead of as a Greek letter.

We thank the reviewer for this helpful point to improve clarity. We would prefer to continue to label this as ϵ in order to be consistent with the convention in density-based clustering, but we have added additional description to the caption to make it clear what this parameter (and the others shown) refer to. Specifically, this reads:

"Single-gene hotspots are computed from the set of cells that highly express a gene, and the fraction of cells within distance ϵ of that cell exceeds density. The hotspot similarity network is constructed with each hotspot as a node, and edges drawn between hotspots whose Jaccard similarity is greater than threshold. Leiden clustering is performed on the hotspot similarity network using resolution parameter resolution. "

* Figure 5a- show the outline of CH6 and CH8 with labels if possible.

Thank you for the suggestion to improve clarity. We have added the outlines and labels for CH6 and CH8 as described.

* "single the score" in methods- I think you mean "since the score"

Thank you – we have corrected this typo.

* "In each permutation, the gene expression vectors of each cell are shuffled..."- are the genes shuffled jointly or is each gene shuffled independently?

The cell labels are shuffled over the count matrix, equivalent to shuffling the genes jointly. We have modified the sentence to read:

"In each permutation, the gene expression vectors of each cell are shuffled across cells (applying the same permutation to each gene), while keeping spatial position the same."

* Methods benchmarking section, last sentence "therefore a total success"- remove the word "total".

We have adjusted the sentence as suggested.

* The software installation instructions are improving but I still got an error when trying to import nest:

`ModuleNotFoundError: No module named 'colorcet'` perhaps this needs to be added to the conda environment file.

We apologize for the issues in importing. The conda environment.yml file does already include the package `colorcet`, and when performing a fresh installation according to the instructions (tested on OSX 13.5 + apple silicon and Ubuntu 22.04) I do not observe the same error, so I am not sure if this is an issue within NeST or if it is some installation issue from another source. I did observe some issues with the installation of Squidpy through pip, and have adjusted the environment.yml file to install it with conda instead. Perhaps that was causing a failure of the pip installation step that led to colorcet not being installed. If this does not address the issue, I would suggest trying to manually install any seemingly missing packages, such as with `pip install colorcet` and see if that fixes it.

Reviewer #1 (Remarks to the Author):

We thank the authors for answering our comments and we are satisfied with their replies.

Reviewer #2 (Remarks to the Author):

Thanks for your hard work on addressing my previous suggestions. I am satisfied with the current manuscript and recommend it move forward to publication.